# LC8 enhances 53BP1 foci through heterogeneous bridging of 53BP1 oligomers

Jesse Howe[1], Douglas Walker[1], Kyle Tengler[2], Maya Sonpatki[1], Patrick N Reardon[3], Justin WC Leung[2], Elisar J Barbar[1]*

[1]Department of Biochemistry and Biophysics, Oregon State University, Corvallis, United States; [2]Department of Radiation Oncology, University of Texas Health and Science Center, San Antonio, United States; [3]Oregon State University NMR Facility, Oregon State University, Corvallis, United States

**Abstract** 53BP1 is a key player in DNA repair and together with BRCA1 regulate selection of DNA double-strand break repair mechanisms. Localization of DNA repair factors to sites of DNA damage by 53BP1 is controlled by its oligomerization domain (OD) and binding to LC8, a hub protein that functions to dimerize >100 clients. Here, we show that 53BP1 OD is a trimer, an unusual finding for LC8 clients which are all dimers or tetramers. As a trimer, 53BP1 forms a heterogeneous mixture of complexes when bound to dimeric LC8, with the largest mass corresponding to a dimer-of-trimers bridged by 3 LC8 dimers. Analytical ultracentrifugation and isothermal titration calorimetry demonstrate that only the second of the three LC8 recognition motifs is necessary for a stable bridged complex. The stability of the bridged complex is tuned by multivalency, binding specificity of the second LC8 site, and the length of the linker separating the LC8 binding domain and OD. 53BP1 mutants deficient in bridged species fail to impact 53BP1 focus formation in human cell culture studies, suggesting that the primary role of LC8 is to bridge 53BP1 trimers, which in turn promotes recruitment of 53BP1 at sites of DNA damage. We propose that the formation of higher-order oligomers of 53BP1 explains how LC8 elicits an improvement in 53BP1 foci and affects the structure and functions of 53BP1.

*For correspondence:
Elisar.Barbar@oregonstate.edu

Competing interest: The authors declare that no competing interests exist.

## Editor's evaluation

This study offers a useful investigation into how 53BP1 and LC8 interact to form higher-order oligomers that are important for DNA repair. The authors provide convincing biochemical and biophysical evidence supporting a model in which LC8 bridges 53BP1 trimers via the QT2 motif. The work establishes a solid foundation for future efforts aimed at elucidating the structural organization and functional relevance of these complexes in vivo, and will be of broad interest to researchers studying DNA damage response and protein complex assembly.

## Introduction

Anti-tumor protein p53 binding protein (53BP1) is a 200 kDa protein involved in DNA repair and cell cycle regulation (*Iwabuchi et al., 1994*; *Rappold et al., 2001*; *Mirza-Aghazadeh-Attari et al., 2019*). Together, breast cancer type 1 susceptibility gene (BRCA1) and 53BP1 regulate pathway selection for DNA double-strand break repair (*Gupta et al., 2014*). 53BP1 binds and protects DNA ends formed by double-strand breaks, limiting end resection and promoting use of nonhomologous end joining (NHEJ) for DNA repair (*Rappold et al., 2001*; *Ward et al., 2003*; *Becker et al., 2018*). BRCA1 is

**Figure 1.** Domain architecture of 53BP1 and preliminary characterization. (**A**) Map of 53BP1 domain architecture showing LC8 binding sites (QT1-3) in blue and the oligomerization domain (OD) in magenta. (**B**) Structure of LC8 dimer (green) bound to two strands of a peptide of partner (blue) (PDB code 5E0L). (**C**) Variable binding motif recognized by LC8. The x-axis shows the residue position, and the y-axis shows the frequency of residues found in known LC8 binding clients. The anchor is the least variable and is assigned residue numbers of –1, 0, and 1. (**D**) Circular dichroism (CD) data for 53BP1 OD (aa 1200–1290) overlaid by BeStSel fit. Data are presented in units of mean residue ellipticity (MRE). Percent secondary structure is reported in F. (**E**) Secondary structure predictions of 53BP1 1200–1290 using JPred, PsiPred, AlphaFold2, and BeStSel fit of CD data. (**F**) Size-exclusion chromatography coupled to multi-angle light scattering (SEC-MALS) of 53BP1 OD (1200–1290) in the loading concentration range of 75–750 µM (on the column 7.5– 75 µM) showing no change in the mass corresponding to a trimer (monomer 13.2, measured 39.2 kDa). (**G**) (Left) Circular dichroism of 53BP1 LC8 binding domain (LBD)-OD overlaid with LBD (1140–1225) spectra, shown here for comparison. In addition to the minima at 200 nm for LBD-OD (1140–1290), indicative of disorder, the minima near 218 nm shows evidence of an ordered region. (Right) Thermal denaturation (red) and refolding (blue) of LBD-OD showing increase in ellipticity at 200 nm. (**H**) SEC-MALS of 53BP1 LBD-OD shows mass consistent with a trimer.

The online version of this article includes the following source data for figure 1:

**Source data 1.** Data for CD and SEC-MALS.

necessary for the eviction of 53BP1 from chromatin, allowing for end resection factors to initiate homology-directed repair (HDR) (*Bouwman et al., 2010*). In the absence of functional BRCA1, the HDR pathway is nonfunctional, and error-prone NHEJ dominates double-strand break repair (*Bunting et al., 2010*). Aberrant NHEJ results in genome instability such as chromosomal rearrangements and radial chromosome formation. Chemotherapy drugs such as PARPi and platinum-based drugs are effective against BRCA-negative cancers (*Rappold et al., 2001*; *Becker et al., 2018*; *Al-Ejeh et al., 2010*; *West et al., 2019*). However, in BRCA-negative cells, inactivation of 53BP1 once again activates HDR, resulting in loss of efficacy of PARPi and other chemotherapy agents. Understanding the functional states of 53BP1 is therefore critical to creating effective drugs and treatment strategies for cancer patients.

In response to DNA damage, 53BP1 forms nuclear bodies which mark sites of DNA double-strand breaks (*Becker et al., 2018*; *Schultz et al., 2000*; *Kilic et al., 2019*). Upon induction of DNA damage, 53BP1 is recruited to sites of DNA double-strand break and is retained on chromatin through interactions between modified histones and the tudor domain (*Zgheib et al., 2009*), ubiquitin-dependent recognition domain (*Fradet-Turcotte et al., 2013*), and oligomerization domain (OD) (*Kelliher et al., 2024*; *Figure 1A*). The long N-terminal disordered domain of 53BP1 (residues 1–1230), which contains 28 putative ATM kinase recognition (S/TQ) motifs, is phosphorylated by ATM kinase at multiple sites (*Zgheib et al., 2005*; *Jowsey et al., 2007*), which results in recruitment of downstream effectors active in protection of DNA ends (e.g. RIF1 [*Zimmermann et al., 2013*], Rev7 [*Ghezraoui et al.,*

2018], and Shieldin [*Noordermeer et al., 2018*]) or in NHEJ (e.g. PTIP [*Wu et al., 2009*]). The recruitment of 53BP1 to nuclear bodies is critical to its function in DNA repair.

53BP1 contains 3 LC8 recognition motifs at the C-terminus of its disordered domain (residues 1150–1230) (*Lo et al., 2005*; *Howe et al., 2022*), followed by the OD (*Zgheib et al., 2009*) (residues 1231–1272), which is poorly characterized. LC8 is a hub protein that binds >100 client proteins at their disordered regions in its two symmetric binding grooves, facilitating the dimerization of the client proteins (*Jespersen et al., 2019a*; *Benison et al., 2007*; *Barbar and Nyarko, 2015*; *Figure 1B*). Interplay between the OD and LC8 binding domain (LBD) is crucial to the function of 53BP1 (*Becker et al., 2018*; *West et al., 2019*). Either the OD mutant (ODm) or LC8 binding mutant (LC8m) reduces 53BP1 nuclear bodies, and a construct containing both mutations reduces nuclear bodies to levels similar to the control (*Becker et al., 2018*). Furthermore, the LC8m shows reduced DNA repair activity, while the ODm is inactive in DNA repair, and the LC8m/ODm construct is not even efficiently recruited to nuclear bodies. The functional synergy between the OD and LC8 binding in the formation of 53BP1 nuclear bodies supports a model in which LC8 either stabilizes dimeric 53BP1 to increase affinity for bivalent interaction with chromatin as proposed (*Becker et al., 2018*) or that LC8 stimulates higher-order oligomerization in 53BP1 (*Lou et al., 2020*).

LC8 binds a highly variable short linear recognition motif in disordered regions of client proteins (*Figure 1C*; *Jespersen et al., 2019a*; *Benison et al., 2007*). A well-conserved triad, TQT, most frequently occupies the anchor of this motif, but the entire sequence is variable. Due to the strong preference for binding to QT motifs, we refer to LC8 binding sites as QT sites (or QTs). Both the OD of 53BP1 and the binding of the three QTs to dimeric LC8 are expected to contribute to oligomerization of 53BP1. Intriguing about our work here is the discovery that 53BP1 is a trimeric LC8 client (*Figure 1H*), leading to the question of how dimeric LC8 binds a trimeric client and the functional role of this binding. The well-described mechanisms through which LC8 regulates oligomeric clients include regulation of the affinity of self-association (*Kidane et al., 2013*; *Barbar and Nyarko, 2014*), structuring of the disordered region (*Nyarko et al., 2013*; *Gaik et al., 2015*), and alignment of domains by restriction of the conformational ensemble of the client (*Jespersen et al., 2019b*). Here, we show a novel binding mode for LC8, bridging of trimeric client subcomplexes, and propose that LC8 improves 53BP1 focus formation through heterogeneous bridging of 53BP1 subcomplexes.

## Results

### Structural characterization of LBD-OD

The OD of 53BP1 spans residues 1231–1279 based on truncation studies (*Zgheib et al., 2009*). Disorder prediction by IUPred (*Erdős et al., 2021*) suggests that residues 1240–1270 are ordered. Since coiled-coils are the most common type of OD in LC8 binding clients (*Barbar and Nyarko, 2014*; *Benison et al., 2008*), we ran predictions for coiled-coil propensity. Neither Waggawagga (*Kollmar and Simm, 2019*) nor Marcoil (*Delorenzi and Speed, 2002*) predicts any coiled-coils in this region. Circular dichroism (CD) of a construct of the OD spanning residues 1200–1290 shows a single minimum near 218 nm (*Figure 1D*) consistent with a primarily beta-sheet structure. Fitting this data with BeStSel (*Micsonai et al., 2015*) estimates a primarily beta-sheet structure with some propensity for an alpha-helical structure (*Figure 1E*). Structure prediction algorithms are in general agreement in the structured regions (*Figure 1E*). AlphaFold2 (*Jumper et al., 2021*) and PsiPred (*McGuffin et al., 2000*) predict an entirely beta-sheet structure, while JPred (*Drozdetskiy et al., 2015*) predicts a mixture of alpha-helices and beta-strands. Since the BeStSel CD-based structure analysis estimates a higher percentage of structure than the sequence-based predictions, we conclude that the increased structure is due to some folding upon oligomerization (*Figure 1E*).

The LBD remains disordered in the context of the OD, as shown by a sharp peak near 200 nm and a smaller minimum around 218 nm in the CD spectrum, consistent with a primarily disordered structure containing some beta-sheet structure (*Figure 1G*). The spectrum of LBD alone (residues 1140–1225) contains only the minimum at 200 nm, indicating the ordered domain of LBD-OD is between residues 1225 and 1290 (*Figure 1G*; *Zgheib et al., 2009*). Thermal denaturation assays of LBD-OD by CD show a decrease in the intensity of the minimum near 215 nm and a simultaneous increase in the minimum at 200 nm (*Figure 1G*). Both LBD-OD (*Figure 1G*) and OD (data not shown) have a $T_m$ at 57°C. Refolding curves show no hysteresis, indicating that both LBD-OD and OD refold reversibly.

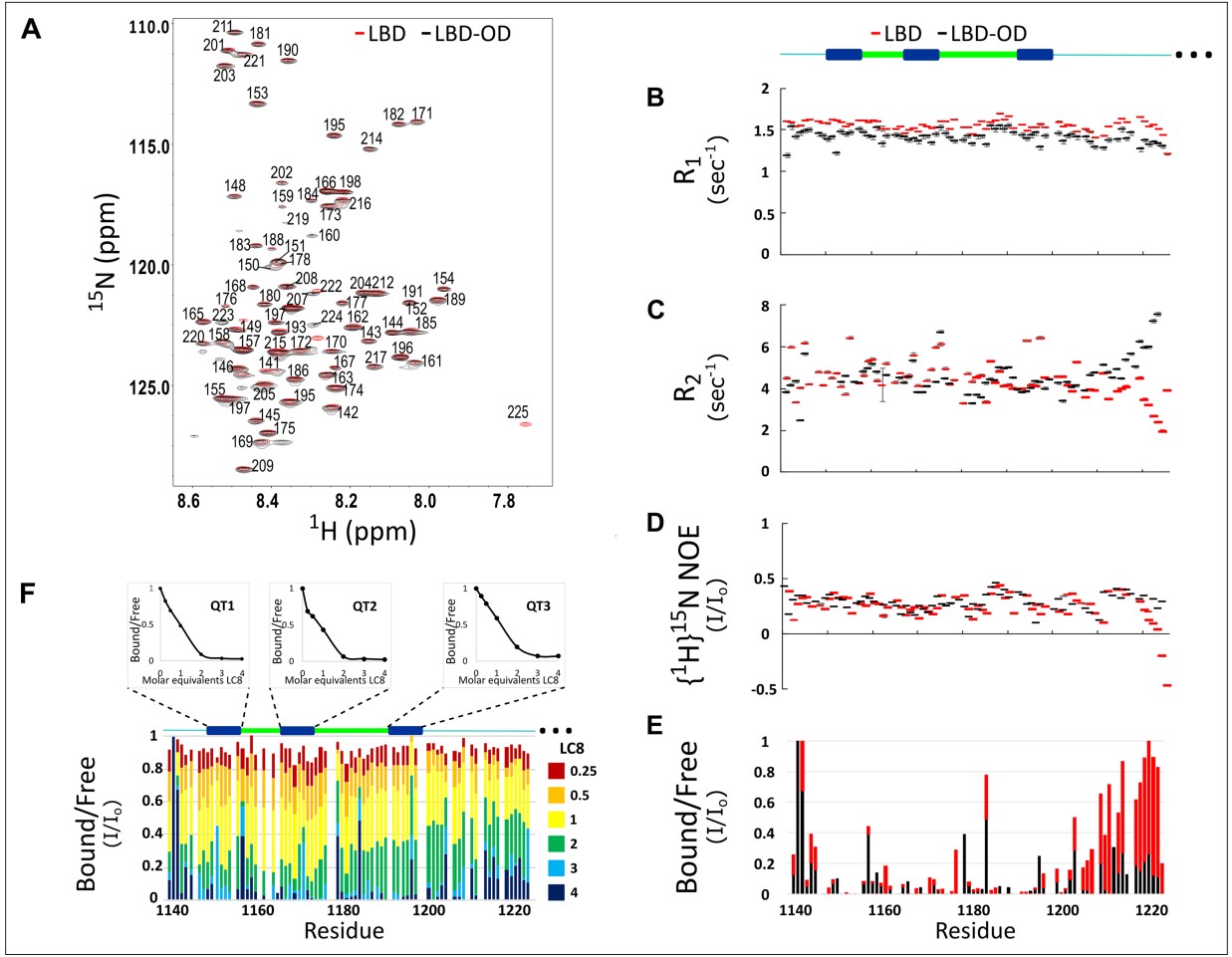

**Figure 2.** Comparison of structure and interactions of LC8 binding domain (LBD) (1140–1225) and LBD-oligomerization domain (OD) (1140–1290) with LC8 probed by NMR. (**A**) Overlay of $^1$H-$^{15}$N heteronuclear single quantum coherence (HSQC) spectra of 53BP1 LBD (red) and LBD-OD (black) acquired at 800 MHz at 10°C in 20 μM sodium phosphate (pH 6.5) and 50 μM sodium chloride with 10% D$_2$O. The peaks are labeled for residues within the 1140-1225 segment without including the one thousands place, which is 1--- for every residue to minimize crowding. Resonances overlay well for almost all residues. (**B**) R1, (**C**) R2, and (**D**) {$^1$H}$^{15}$N NOE overlay of LBD (red) and LBD-OD (black). Diagram of 53BP1 LBD with LC8 QT sites labeled in blue shown at the top. (**E**) Overlay of $^{15}$N LBD and $^{15}$N LBD-OD peak height in $^1$H-$^{15}$N HSQC in the presence of 4 molar equivalents of LC8. (**F**) Titration of uniformly labeled $^{15}$N LBD-OD (1140–1290) with unlabeled LC8. A diagram of LBD-OD is shown above the plot which highlights the locations of QT sites (blue). The average signal for the 8 residues comprising the QT site is shown in decay curves above each QT site as a function of LC8 concentration. Note that since OD signals are missing from the spectra, there are no measurements reported in B–F in this region.

The online version of this article includes the following source data for figure 2:

**Source data 1.** NMR relaxation and titration data.

Size-exclusion chromatography coupled to multi-angle light scattering (SEC-MALS) of the OD (1200–1290) shows a mass of 39.2±0.6 kDa as an average of 4 runs loaded at concentrations between 75 and 750 μM (dilution over column is ~10×) (*Figure 1F*), which suggests a trimeric form dominates in this concentration range (expected trimer mass of 39.5 kDa) and that the OD is a stable trimer. LBD-OD shows a single peak with a mass of 50 kDa (*Figure 1H*). Since the calculated mass expected for a monomeric LBD-OD is 17.0 kDa, the mass measured by SEC-MALS suggests that LBD-OD is a trimer (expected mass of 51 kDa for trimer). To date, all known oligomeric clients of LC8 are either dimers or tetramers, and thus the mechanism for complex formation between a dimer and a trimer is not known.

## Disorder of the LBD is retained in the LBD-OD

The $^1$H-$^{15}$N heteronuclear single quantum coherence (HSQC) of LBD-OD shows low peak dispersion and overlays well with the HSQC of LBD (*Figure 2A*). No additional peaks for the OD were detected

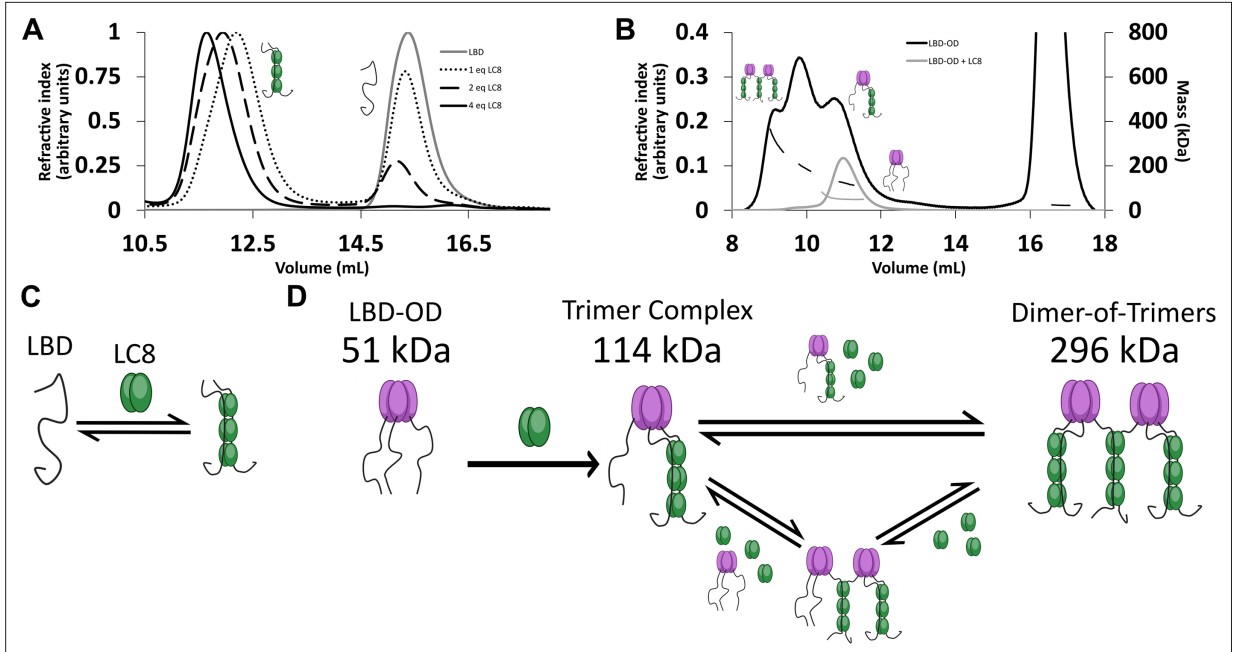

**Figure 3.** LC8 binding domain (LBD)-oligomerization domain (OD) (1140–1290) forms a heterogeneous complex when bound to LC8. (**A**) Titration of 53BP1 LBD (1140–1225) results in a single species with large mass, even at low stoichiometric equivalents. Data is adapted from *Howe et al., 2022*, and shown here for comparison. (**B**) Size-exclusion chromatography coupled to multi-angle light scattering (SEC-MALS) of free LBD-OD and LBD-OD with 4 molar equivalents of LC8 shows the formation of a heterogeneous mixture of complexes with masses in the 120–296 kDa range (*Table 2*). (**C**) Model of LBD binding LC8. A duplex containing 2 strands of LBD and 3 dimers of LC8 is created in a single step. (**D**) Model of LBD-OD binding LC8. 53BP1 trimers bind LC8, forming a trimer complex containing 1 53BP1 trimer and 3 LC8 dimers. The trimer complex binds excess LC8-forming bridged complexes. We simplify our model to show trimer and dimer-of-trimers only, as these are the major species based on the 53BP1 showing no concentration-dependent shift from a trimeric mass analyzed by SEC-MALS.

The online version of this article includes the following source data for figure 3:

**Source data 1.** Titration data from SEC-MALS.

in the concentration range of 50–400 µM and in the temperature range of 10–40°C. Cα-Cβ chemical shift indexing of LBD and LBD-OD shows shifts less than 0.5 for most residues, indicating that both constructs are disordered between residues 1140 and 1225 (data not shown). Most residues of the LBD in the context of the OD show similar dynamic behavior to that of the LBD alone, indicating that the LBD remains disordered in the LBD-OD construct (*Figure 2B–E*). However, residues 1210–1225, which are at the end of the LBD, appear less dynamic with high $R_2$ values (*Figure 2C*) compared to those for the same residues in the LBD alone, as expected due to its proximity to a self-associated domain. Taken together, these data indicate that the LBD-OD retains flexibility in the intrinsically disordered LBD but possesses a folded region that corresponds to the OD.

**Table 1.** Masses measured by size-exclusion chromatography coupled to multi-angle light scattering (SEC-MALS).

**Measured**

| Construct | High mass (kDa) | Intermediate mass (kDa) | Low mass (kDa) |
|---|---|---|---|
| LBD-OD | 296±2.5 | 190±1.7 | 120±2.2 |
| QT2-OD | 168±1.2 | 104±0.6 | 80±.5 |
| QT3-OD | – | – | 69±0.4 |
| QT12-OD | 247±1.2 | 178±0.8 | 112±0.5 |
| QT13-OD | – | – | 92±0.5 |
| QT23-OD | 233±1.3 | 123±0.7 | 101±0.6 |

**Table 2.** Expected masses of LC8 binding domain (LBD)-oligomerization domain (OD) mutants in different LC8 complexes.

| Complex | Bridged complex ratio (53BP1 trimers:LC8 dimers) | Expected mass (kDa) | Trimer complex (53BP1 trimers:LC8 dimers) | Expected mass (kDa) |
| --- | --- | --- | --- | --- |
| LBD-OD | 2:9 | 292.8 | 1:3 | 114.6 |
| One-site mutants | 2:3 | 165.6 | 1:1 | 72.2 |
| Two-site mutants | 2:6 | 229.2 | 1:2 | 93.4 |

Titration of LBD with LC8 shows uniform signal attenuation in residues 1150–1200 of 53BP1, suggesting no binding preference for any particular site (*Howe et al., 2022*). Titration of LBD-OD with LC8 shows similar uniform signal attenuation for the same residues (*Figure 2F*), suggesting that the cooperative binding of LBD is conserved in the presence of the OD.

## 53BP1 OD forms heterogeneous complexes when bound to LC8

The interaction between LBD and LC8 by analytical SEC and NMR supports a model in which LBD binds LC8 cooperatively in a single step (*Howe et al., 2022*). SEC is shown here for comparison (*Figure 3A*). In contrast, SEC-MALS of LBD-OD with excess LC8 shows a heterogeneous complex containing at least three major populations (*Figure 3B*) with masses in the 120–296 kDa range (*Table 1*). Dimeric LBD-OD with 3 LC8 dimers has a calculated mass of 95 kDa, much smaller than what we observed. A complex containing 1 LBD-OD trimer and 3 LC8 dimers is expected to have a mass of 114 kDa, and a complex containing 2 LBD-OD trimers and 9 LC8 dimers would have a mass of 293 KDa (*Table 2*), which are both consistent with the measured masses. Potential intermediates that contain 2 LBD-OD trimers could occupy between 3 and 9 dimers of LC8. We cannot rule out the presence of dimers and tetramers of 53BP1 bound to LC8, but due to the lack of a concentration-dependent shift in OD oligomerization as seen in SEC-MALS (*Figure 1F*), we simplify our model to show the expected major populations of 53BP1 trimers. Our SEC-MALS suggests that LBD-OD forms a trimeric complex with 3 dimers of LC8 and a bridged complex containing 2 trimers of LBD-OD and up to 9 dimers of LC8 (*Figure 3D*). For comparison, the LBD alone shows a much simpler titration behavior (*Figure 3A*) with a single peak for the LBD-LC8 complex implying a single step to form a duplex complex (*Figure 3C*).

## Structural basis of heterogeneity in LBD-OD complexes with LC8

To determine the importance of multivalency in 53BP1-LC8 interactions, we generated a series of mutants which change the three anchor residues of the LC8 binding motif to alanine (AAA), eliminating LC8 binding at the sites of the mutations. We name each construct by the sites left intact (e.g. QT13-OD has AAA mutation in QT2, leaving QTs 1 and 3 to bind LC8) (*Figure 4A*).

In sedimentation velocity analytical ultracentrifugation (SV-AUC) of one-site LBD-ODm (*Figure 4B*), two peaks were observed for QT2-OD, indicating a mixture of different size complexes, while QT3-OD primarily formed a single larger complex. All the two-site QT variants (*Figure 4B*) show multiple complexes in excess LC8 supporting the presence of an activation barrier between the different size complexes, which differs depending on the QT variants used.

SEC-MALS analysis identified the mutants that form the most stable bridged complexes with LC8 that survived dilution on the SEC column (*Figure 4C*). In contrast to SV-AUC, QT3-OD is exclusively a single complex, while QT2-OD retains a small fraction of a larger presumably bridged complex. QT1-OD is fully dissociated, and so we do not include it in our analysis. This difference between SEC and SV-AUC profiles suggests that while both QT2 and QT3 mutants can form bridged complexes with excess LC8 as observed by SV-AUC, only QT2-OD maintains stable bridging upon dilution during SEC. Analysis of the two-site QT variants further demonstrated that complexes containing intact QT2 have populations with larger masses consistent with the bridged complex. These findings suggest that QT2 is necessary to form stable intermolecular interactions resulting in intact bridged complexes.

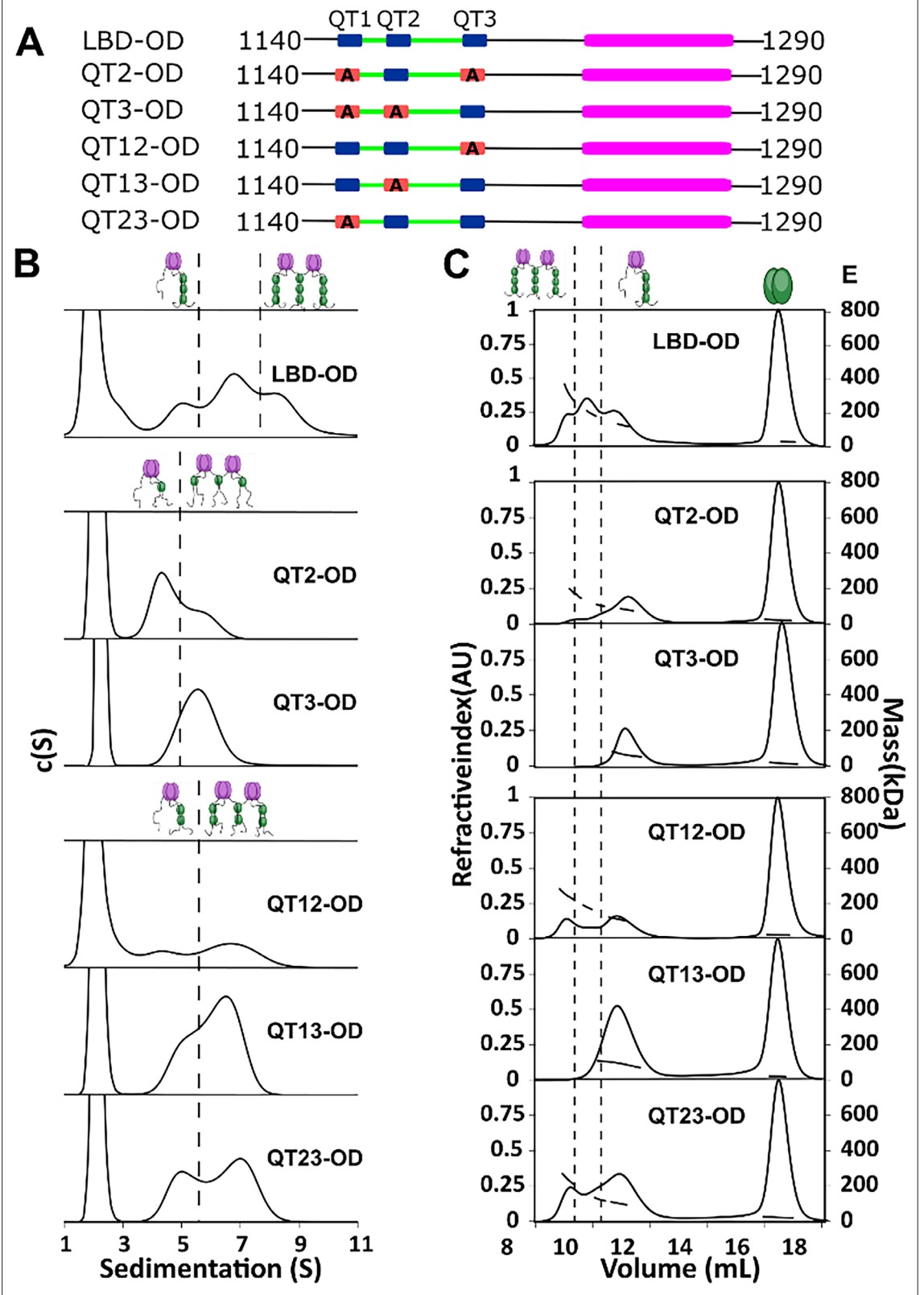

**Figure 4.** QT2 determines the stability of bridged complex. (**A**) Domain maps of LC8 binding domain (LBD)-oligomerization domain (OD) QT mutants. LC8 binding sites (QT) are shown in blue, while those mutated to abolish binding are shown in red. (**B**) Sedimentation velocity analytical ultracentrifugation (SV-AUC) of wild-type (WT) (top) one-site mutants (center) and two-site mutants (bottom) in 50 mM sodium phosphate and 150 mM sodium chloride at 25°C. Dotted lines separate large mass peaks from small mass peaks. Peaks with high S value are assigned to the dimer-of-

*Figure 4 continued on next page*

*Figure 4 continued*

trimers complex and low mass peaks to the trimer complex, represented in cartoon. (**C**) SEC-MALS of complexes formed by LBD-OD and mutants in 50 mM sodium phosphate and 150 mM sodium chloride (pH 7.2). Dotted lines separate large mass peaks from small mass peak. The dimer-of-trimers complexes are only observed for the complexes containing intact QT2.

The online version of this article includes the following source data for figure 4:

**Source data 1.** Titration data from SEC-MALS and AUC.

## Thermodynamics analysis of LBD-OD:LC8 interactions reveals a unique role for QT2

Isothermal titration calorimetry (ITC) showed different binding affinities for each site in the LBD (*Howe et al., 2022*). Importantly, QT2 is the only site with an unfavorable entropy, and QT1 is the weakest binding site overall. Here, we investigate LC8 binding to 53BP1 in the context of the OD.

Analysis of isotherms from LC8 titrated into LBD-OD and the one- and two-site mutants with the one-set-of-sites (OSS) model shows enthalpically favorable interactions with $K_d$ in the sub-micromolar to low micromolar range (*Figure 5K*, *Table 3*). The wild-type (WT) LBD-OD (*Figure 5A*) shows a stoichiometry near 3:1 (3 LC8:1 53BP1; or 3 LC8 dimers and 1 trimer of 53BP1; or 9 LC8 dimers and 2 trimers of 53BP1). The latter suggests that a bridged complex could be formed when all the LC8 sites are filled. However, ITC alone is not sufficient to differentiate between all possible models.

The one-site LBD-ODm (*Figure 5B–E*) have similar binding properties to LBD (*Howe et al., 2022*) in that QT2 is the only one-site mutant with unfavorable entropy. Interestingly, both QT1 and QT3 binding give non-sigmoidal isotherms that we attribute to multistep binding. One model that explains the data is that two chains of the trimer bind to LC8, leaving the third chain available to bridge the available chain from another trimer (*Figure 5J*). To test this model, we reversed the direction of titration, resulting in an excess of LC8 in early titration points, which we hypothesized would push the reaction such that each QT3-OD binding site was fully bound at every injection. Titrating QT3-OD into LC8 produced a perfectly sigmoidal isotherm, indicating that no intermediate is formed in line with our model (*Figure 5E*).

The resulting thermodynamic parameters as calculated by the OSS model are based on a convolution of the stronger intratrimeric binding and the weaker intertrimeric (bridging) LC8 binding and therefore are a weighted average of the two events. The sigmoidal isotherm observed when titrating LC8 into QT2-OD can be explained as due to the intermediate being only briefly occupied, suggesting that the bridging interaction has a stronger binding affinity similar to that of the intratrimeric LC8 binding than with the other single site mutants. We modeled our system as a hexamer with three binding events (two intratrimer events and one bridging event) using a subsequent binding sites (SBS) model in Origin 7.0 (*Figure 5L*), which enables extracting thermodynamics of multiple binding events. Using this model, we identify two clearly different binding affinities in the one-site LBD-OD variants. We assign the lower affinity binding event to the bridging of 53BP1 trimers and the higher affinity to the intratrimer interactions. For the intratrimer interactions, the difference in affinity for each variant is very small ($K_d$ between 0.3 and 0.6 µM), while the difference between affinities for bridging interactions is larger ($K_d$ between 1 and 5 µM). The variant with the smallest difference between bridging affinity and intratrimer interactions is QT2-OD (intratrimer $K_d$ = 0.3 µM, intertrimer $K_d$ = 1.1 µM). This small difference explains the absence of intermediates in ITC isotherms for LBD-OD containing intact QT2. Additionally, the fact that QT2-OD has the strongest affinity for the bridging interaction provides a mechanistic explanation for our observation that QT2 is essential for a strongly bridged dimer-of-trimers complex. Overall, the SBS fitting of LBD-OD one-site mutant isotherms clearly shows that there are multiple complexes formed with different populations, that the binding at the three recognition motifs is different, and that binding at QT2 is the only binding with a low energy barrier between the different complexes.

To better understand the interplay between multiple LC8 sites in 53BP1, we performed ITC on two-site mutants of LBD-OD (*Figure 5F–I*). All interactions have affinities around 0.5 µM, but QT13-OD, the only two-site mutant not containing QT2, gives a non-sigmoidal isotherm. Again, when we reverse the direction of titration (QT13-OD into LC8), the isotherm becomes sigmoidal (*Figure 5H and I*). Since all LBD-ODm containing QT2 have sigmoidal isotherms, we attribute the non-sigmoidal

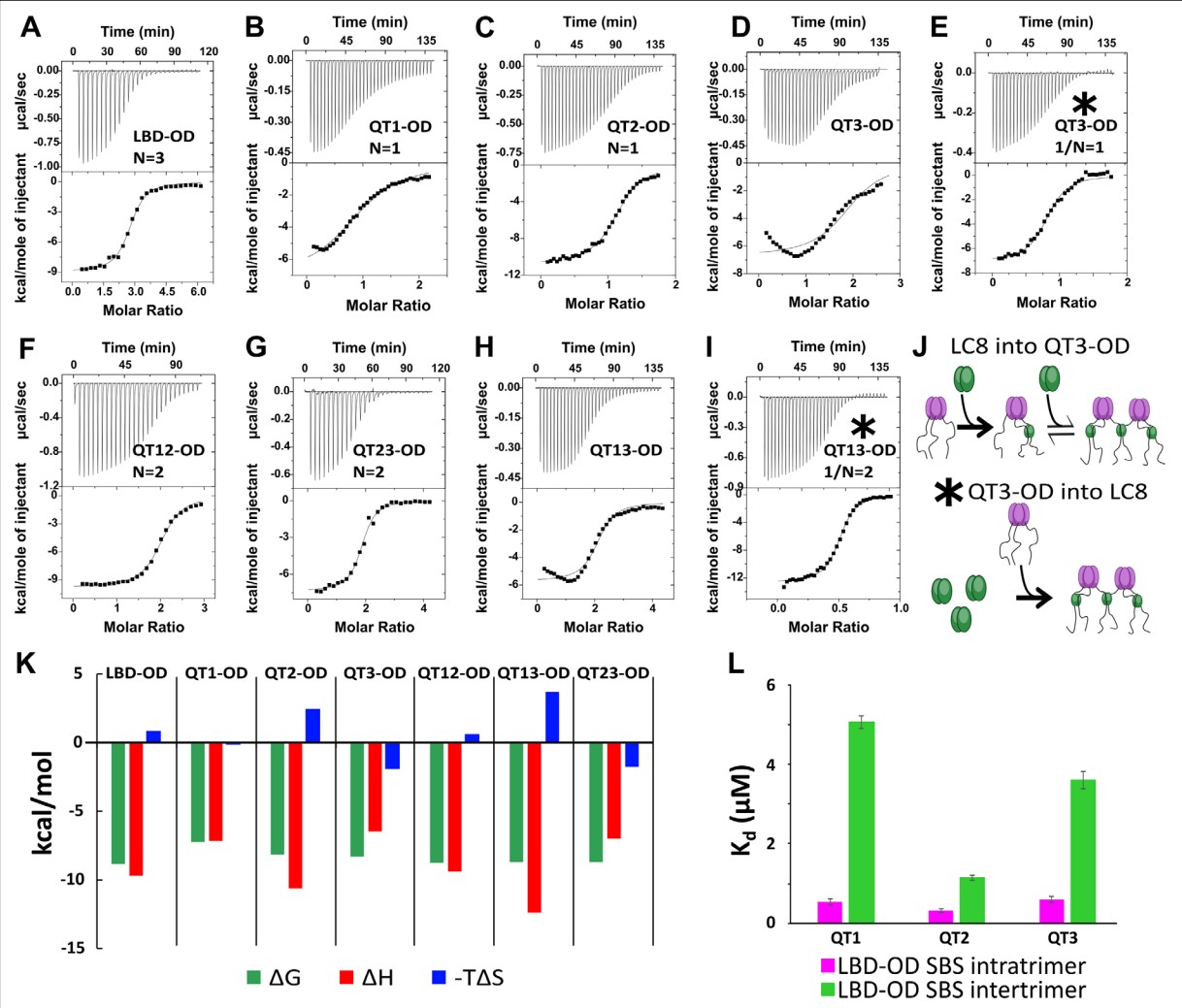

**Figure 5.** Thermodynamics of interactions of LC8 binding domain (LBD)-oligomerization domain (OD) and mutants with LC8. (**A–D, F–H**) Representative isotherms for LBD-OD (**A**), QT1-OD (**B**), QT2-OD (**C**), QT3-OD (**D**), QT12-OD (**F**), QT13-OD (**G**), and QT23-OD (**H**) with LC8 in 50 mM sodium phosphate and 150 mM sodium chloride (pH 7.2) at 25°C. In these experiments, LC8 at 250–400 µM was titrated into LBD-OD or mutant at 10–30 µM. (**E, I**) Isotherm of QT3-OD (**E**) and QT13-OD (**I**) at 200 µM titrated into 20 µM LC8. The downward inflection early in the titration is not present in this isotherm, as it is in the titration of LC8 into mutant in D and H. (**J**) A cartoon illustrating how a stable intermediate is formed during titration. (Top) When LC8 is titrated into QT3-OD, excess 53BP1 binds LC8, resulting in forming the trimeric intermediate. (Bottom) When QT3-OD is titrated into LC8, excess LC8 allows the bridged complex to form without a stable intermediate. (**K**) Bar graph of thermodynamic parameters for A-G isotherms fit to one-set-of-sites (OSS) model. Thermodynamic parameters are shown in *Table 1*. (**L**) Affinities of one-site binding in 53BP1 determined from fits to the OSS model for LBD and LBD-OD are in gray and orange and using subsequent binding sites (SBS) model in cyan (intratrimer) and magenta (bridging). QT2-OD has the smallest difference in affinity between intratrimer and bridging interactions. For statistical analysis: QT1 n=3, QT2 n=4, QT3 n=5. Error bars are defined as the aggregated errors of each individual Origin fit line. The method for aggregating the different values and their errors is in the Methods section.

isotherms to the formation of a trimeric intermediate and suggest that QT2 binding may pay the entropic penalty for bridging 53BP1 trimers.

## Reducing linker length between LBD and OD stabilizes dimer-of-trimers

For ASCIZ, which binds LC8 multivalently to regulate transcription, linker length contributes to both compositional (*Walker et al., 2023*) and conformational (*Reardon et al., 2020*) heterogeneity. To determine a possible similar effect for linker length on 53BP1 heterogeneity, we focused on the linker between 53BP1 LBD and OD, hypothesizing that the LBD-OD linker would regulate the bridging interaction. We generated two mutants of LBD-OD by deleting 5 and 15 residues (Δ1206–1210 and Δ1206–1220, respectively) (*Figure 6A*) in the ~30 residue-long linker. Shortening the linker by 5

**Table 3.** Thermodynamics of LBD-OD:LC8 interactions measured by **i**sothermal titration calorimetry (ITC).

| Construct | N | $K_d$ (µM) | ΔH (kcal/mol) | –TΔS (kcal/mol) | ΔG (kcal/mol) |
|---|---|---|---|---|---|
| LBD-OD | 2.8±0.4 | 0.4±0.1 | –9.7±2 | 0.9±2.3 | –8.8±0.2 |
| QT1-OD | 1.1±0.04 | 5.0±1 | –7.1±0.4 | –0.1±0.6 | –7.2±0.2 |
| QT2-OD | 1.2±0.02 | 1.2±0.03 | –10.6±0.3 | 2.5±0.3 | –8.1±0.01 |
| QT3-OD[*] | 0.87±0.04 | 0.86±0.04 | –6.9±0.1 | –1.4±0.1 | –8.3±0.03 |
| QT12-OD | 2.3±0.1 | 0.5±0.3 | –9.4±0.5 | 0.7±0.9 | –8.7±0.4 |
| QT13-OD[*] | 2.1±0.1 | 0.7±0.02 | –5.6±0.1 | –2.8±0.1 | –8.4±0.02 |
| QT23-OD | 1.9±0.05 | 0.4±0.1 | –7.0±0.3 | –1.7±0.4 | –8.7±0.2 |

[*] Values extracted from reverse experiment (LC8 binding domain [LBD]-oligomerization domain [OD] mutant titrated into LC8). Stoichiometries are presented as reciprocal for these experiments for simple comparison to other mutants.

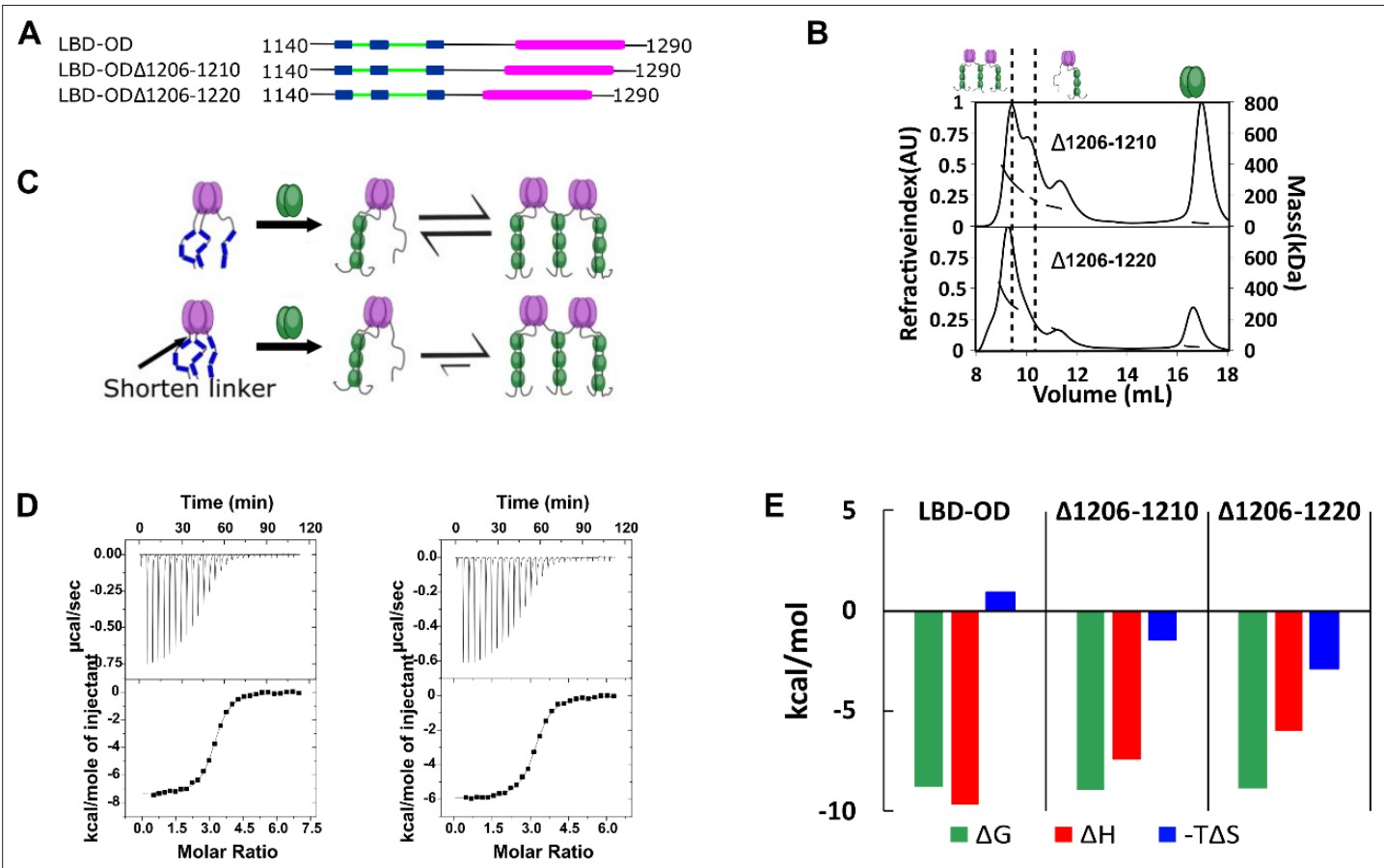

**Figure 6.** Linker length between LC8 binding domain (LBD) and oligomerization domain (OD) determines the stability of bridged complex. (**A**) Domain map of 53BP1 linker deletion mutants. (**B**) Size-exclusion chromatography coupled to multi-angle light scattering (SEC-MALS) of 53BP1 linker deletion mutants shows a high proportion of bridged complexes. (**C**) Model of the effect of reducing the linker in 53BP1 LBD-OD. Shorter linker results in increased population of bridged complex seen by SEC-MALS. (**D**) Isothermal titration calorimetry (ITC) shows sigmoidal curves with stoichiometry near 3, consistent with a single binding step. (**E**) The entropic contribution becomes more favorable as the linker length is decreased, but the overall affinity of the interaction is unchanged.

The online version of this article includes the following source data for figure 6:

**Source data 1.** SEC-MALS data.

residues results in a population shift to a larger presumably bridged complex, and further reducing the length by 15 residues shifts the population to an almost completely larger bridged complex (*Figure 6B*), suggesting that shortening the LBD-OD linker stabilizes the dimer-of-trimers complex (*Figure 6C*). In support of this model, ITC shows sigmoidal isotherms for both linker deletion mutants (*Figure 6D*). The entropic contribution becomes more favorable as the linker length is decreased, while the overall affinity of the interaction is unchanged (*Figure 6E*).

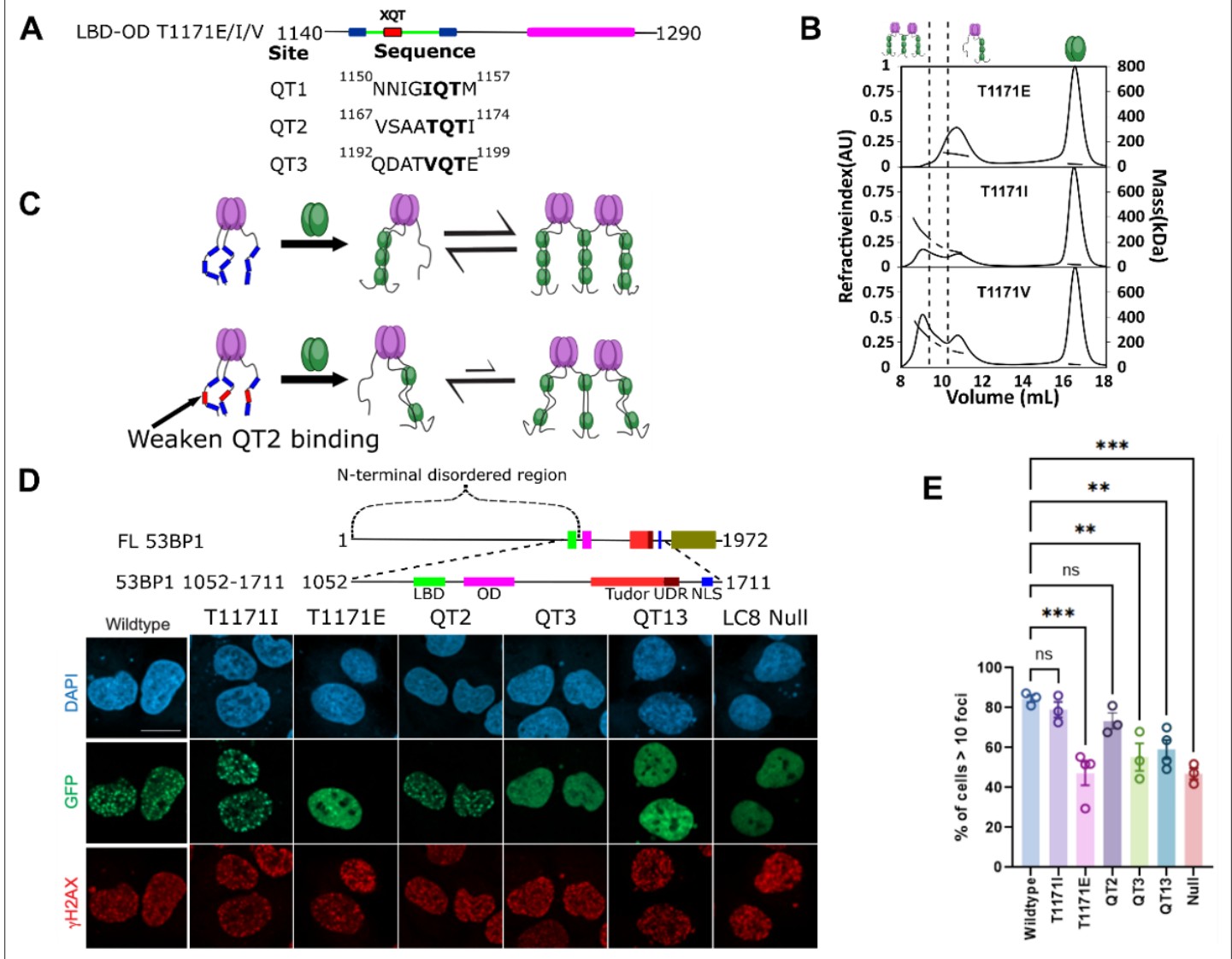

**Figure 7.** LC8 bridging promotes 53BP1 recruitment to DNA damage sites. (**A**) (Top) Domain map of 53BP1 QT2 mutants. (Bottom) Sequence of the three QT sites (QTs) in 53BP1. QT2 is the only site with the canonical TQT anchor. (**B**) Mutation of the −1 site (T1171) in 53BP1 has drastic effects on the relative population of trimer and bridged complexes. The phosphomimetic (T1771E) elutes completely as a trimer, while both phosphodeficient mutants (T1171I/V) generate a mixture of trimer and bridged complex. (**C**) Model showing the effect of reducing the linker length in 53BP1 LC8 binding domain (LBD)-oligomerization domain (OD) on the proportion of bridged complex. (**D**) (Top) Domain map of constructs transfected into U2OS cells. These constructs span residues 1052–1711 and contain the LBD, OD, Tudor, UDR, and nuclear localization sequence (NLS) of 53BP1. (Bottom) Representative immunofluorescence micrographs of GFP-labeled 53BP1 fragment aa1052–1711 wild-type and mutants, at 1 hr after 10 Gy irradiation. Scale bar is 20 μm (**E**) Quantification of 53BP1 foci as presented in D for the indicated expression vectors. The error bars correspond to mean ± SEM. The total number of cell analyzed is >180. The total number for each group is the following: WT 184, T1171I 200, T1171E 259, QT2 292, QT12 217, QT13 209, LC8 Null 250. One way ANOVA was used for statistical analysis. ns- non significant; **<0.01; ***<0.001.

The online version of this article includes the following source data for figure 7:

**Source data 1.** SEC-MALS data.

## QT2 sequence determines the proportion of bridged complex

The sequence specificity of LC8 binding has been studied in great detail, revealing a variable motif which binds LC8 with varying affinities depending on the sequence of the client (*Jespersen et al., 2019a*; *Benison et al., 2007*; *Benison et al., 2008*; *Erdős et al., 2017*; *Clark et al., 2016*). Most important to the affinity of LC8 binding are three anchor residues within the LC8 binding motif (*Figure 7A*). Mutations within the LC8 binding anchor have large impacts on binding, but the –1 position exhibits the most variability (*Figure 1B*). Within 53BP1, all three QTs contain a different residue in the –1 position (*Figure 7A*, bottom). Two LBD-OD QT2 variants which are still expected to bind LC8 (T1171I and T1171V), and a phosphomimetic variant which is expected to greatly reduce binding for QT2 (T1171E) and alter the populations of LBD-OD complexes observed in SEC-MALS (*Figure 7B*). As expected, T1171E elutes completely as a trimer complex, like QT13-OD, while both T117I and T1171V form a mixture of complexes, presumably a trimer and bridged complexes. T1171I forms a similar population of trimer and bridged complex to the WT, while T1171V forms a significantly higher population of the larger bridged complex (*Figure 7B*), indicating that sequence specificity of QT2 is a major determinant of 53BP1 heterogeneity. BLAST analysis of the human 53BP1 sequence shows that the anchor of QT2 is well conserved in animals, in support of a critical role for 53BP1-LC8 interactions being tuned to maintain heterogeneity of 53BP1 oligomerization.

## Formation of a bridged complex is correlated with enhancement in 53BP1 focus formation

53BP1 contains a minimal focus forming (MFF) region spanning residues 1220–1711, which includes the OD, tandem tudor domain, ubiquitin-dependent recruitment domain, and nuclear localization sequence (*Zgheib et al., 2009*; *Figure 7D*). While LC8 binding is not required for focus formation (*Kilic et al., 2019*), it improves focus formation and recruitment of 53BP1 to chromatin (*Becker et al., 2018*; *West et al., 2019*). A series of 53BP1 mutants spanning the LBD and MFF were tested for their ability to improve 53BP1 focus formation in response to ionizing radiation. The phosphodeficient mutant (T1171I) produces an enhancement in 53BP1 foci similar to the WT protein (79 ± 7% for T1171I vs 85 ± 3% for WT), while the phosphomimetic shows no significant difference in the number of cells with >10 foci over a mutant with all three QTs mutated (47 ± 12% for T1171E vs 47 ± 5% for LC8 binding null) (*Figure 7D and E*). The similarity of the phosphodeficient mutant to the WT and the strong loss of foci in the phosphomimetic show that QT2 is not phosphorylated in response to ionizing radiation and argues against the importance of phosphorylation at this site in nuclear repair foci. Constructs with QT2 binding removed by AAA mutation (QT3 and QT13) fail to show a significant improvement in focus formation over the LC8 binding null mutant, while the mutant containing only QT2 does provide an enhancement. All of this together suggest a role for QT2 in regulating the enhancement from LC8 in focus formation of 53BP1, which we propose to be mediated through heterogeneous bridging of 53BP1 trimers.

## Discussion

53BP1 is a key player in the regulation of DNA damage repair, entering DNA repair foci upon induction of DNA damage and orchestrating selection of double-strand break repair pathway (*Bunting et al., 2010*; *Schultz et al., 2000*; *Lottersberger et al., 2013*). While the OD of 53BP1 is required for accumulation of 53BP1 in DNA repair foci, LC8 binding rescues some 53BP1 foci in the absence of the OD (*West et al., 2019*; *Kilic et al., 2019*). Mutation of LC8 binding sites results in DNA repair-deficient 53BP1 and reduces the number of nuclear bodies formed upon induction of DNA damage, suggesting synergistic function between the 53BP1 OD and LC8 (*Becker et al., 2018*). Since LC8 functions as a dimerization hub, often regulating the dimerization of client proteins or contributing to higher-order oligomerization (*Kidane et al., 2013*; *Barbar and Nyarko, 2014*; *Barbar, 2008*), it was proposed that LC8 rescues 53BP1 foci by dimerizing 53BP1 in the absence of the OD. Here, we provide the first evidence that 53BP1 is a trimer, not a dimer, that its interaction with LC8 forms a heterogeneous mixture containing bridged complexes, and that this bridging of two trimers by LC8 is what regulates 53BP1 foci formation.

## 53BP1 is a trimer and forms a dimer-of-trimers with LC8

While clients of LC8 are often dimers (*Barbar and Nyarko, 2014*; *Gaik et al., 2015*; *Jespersen et al., 2019b*), and occasionally tetramers (*Szaniszló et al., 2022*; *Rodriguez Galvan et al., 2021*), there are no reports of trimeric clients. Interestingly, we find that 53BP1 is a trimer and that LC8 can stimulate higher-order oligomerization by bridging two trimers of 53BP1 (*Figure 3B and D*). The trimer association state is confirmed by peak and mass similarity in SEC-MALS over a 10-fold difference in concentration. It is also confirmed by using two constructs containing the OD but of different lengths and both giving a mass for a trimer, indicating that self-association is not dependent on the construct used.

How does the 53BP1 trimer bind to the dimeric LC8 is a complicated question and requires rigorous investigation. A titration of 53BP1 LBD-OD analyzed by $^1$H-$^{15}$N HSQC shows uniform peak loss at the three LC8 binding sites (*Figure 2F*), suggesting that LC8 binds 53BP1 cooperatively and that there is no preference for any QT to be filled before other QTs. Results from AUC show heterogeneous complexes, while SEC-MALS identifies the multiple complexes that survived dissociation on the column. Multiple peaks from both measurements are assigned to species with masses corresponding to LBD-OD bound to LC8 in trimer, bridged dimer-of-trimers, and intermediate species that could be dimeric, trimeric, or tetrameric (*Figure 3B*). The difference in complexes observed by SEC-MALS and AUC indicates that while all QTs can generate multiple complexes, only QT2 is effective at forming the larger presumably bridged species stable enough to remain bound upon dilution during SEC (*Figure 4B and C*).

ITC supports a model in which QT2 is the primary motif that LC8 uses to form the larger complexes (*Figure 5*). When titrating LC8 into LBD-ODm that retain a single LC8 site, QT2 is the one-site mutant that has the highest favorable enthalpy for LC8 binding (*Figure 5K*); this is observed with both LBD and LBD-OD, indicating it is a property of the motif. Additionally, all LBD-ODm with QT2 binding removed show isotherms consistent with two-step binding. These isotherm shapes are not seen in titrations of LBD alone. For example, QT13-OD would first form a stable trimer bound to 2 LC8 dimers, leaving one of the LBD chains completely unoccupied, which upon further addition of LC8 will form a bridged trimer. When the titration is reversed (QT13-OD into LC8), there is no intermediate detected because the trimer complex is not stable in excess LC8, providing support for our model (*Figure 5J*). SBS fits of ITC data of one-site LBD-ODm show a small difference in affinity between bridging and intratrimer interactions only for QT2 (*Figure 5L*), which explains the absence of an intermediate species in constructs containing QT2.

Together, these data suggest a complex structure in which the 53BP1 OD trimers are bridged by LC8 and stabilized through interaction with QT2.

## LC8 binding forms heterogeneous 53BP1 complexes

LC8 is a hub protein involved in binding >100 client proteins at disordered regions (*Jespersen et al., 2019a*; *Barbar and Nyarko, 2015*), facilitating their dimerization and regulating their physiological functions. While it is clearly established that binding to LC8 results in dimerization of the client (*Barbar, 2008*), the structural and physiological consequences of LC8 binding can be more complex. Rigidification of the disordered strand (Nup159) (*Nyarko et al., 2013*), restriction of conformational ensemble (RavP) (*Jespersen et al., 2019b*), regulation of self-association (Swa) (*Kidane et al., 2013*), higher-order oligomerization (LCA5) (*Szaniszló et al., 2022*), and client regulation through cooperative binding (ASCIZ) (*Rapali et al., 2011*; *Clark et al., 2018*) are all mechanisms by which LC8 regulates its clients' functions. Here, we identify higher-order assembly of trimeric 53BP1 as a novel binding mode for LC8.

The outcome of the interaction between 53BP1 and LC8 is higher-order oligomerization of 53BP1, but there is an upper boundary to the size of the complex formed. Once 53BP1 forms the bridged dimer-of-trimers complex, there are no LBD chains available to bind (trimer 53BP1 leaves one free chain after LC8 binds) (*Figure 3D*). A similar mechanism is seen in the interaction between LCA5 and LC8 (*Szaniszló et al., 2022*). LCA5 has two LC8 binding sites and tetrameric coiled-coils; however, the arrangement of these domains is unique for LC8 clients. The two coiled-coils are located on either side of the LC8 binding sites, and as a result, each LC8 site is capable of bridging to a different LCA5 tetramer. Since there is always a free LC8 site to bridge to another LCA5 subcomplex, the higher-order oligomerization is unbounded, resulting in large assemblies of LCA5-LC8 that appear as beads-on-a-string by electron microscopy. While LCA5 and 53BP1 both form higher-order assemblies

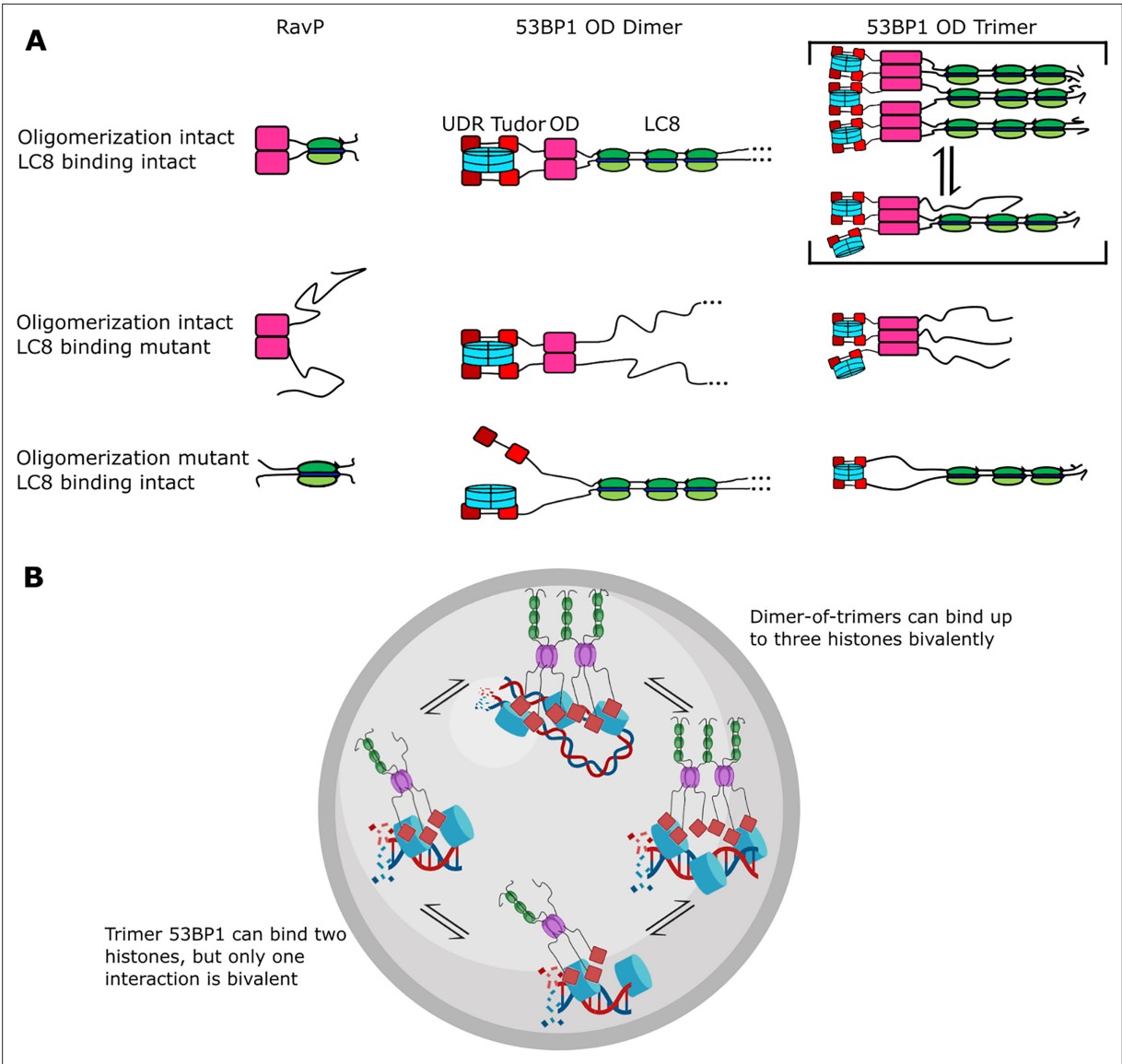

**Figure 8.** Mechanism of LC8 enhancement of 53BP1 focus formation. (**A**) Cartoon showing roles of LC8 in an interaction with RavP (*Jespersen et al., 2019b*) (left), published mechanisms for LC8 with 53BP1 (*Becker et al., 2018*) (center), and revised mechanism based on this work (right). In RavP, the N-termini are aligned with LC8, but when LC8 binding is removed, the termini occupy a larger conformational ensemble. In the dimeric 53BP1 oligomerization domain (OD) model, 53BP1 is a dimer as long as either the OD or LC8 binding is intact. This model does not fully capture the effect of OD and LC8 mutation of 53BP1 focus formation and cannot be correct because 53BP1 is a trimer, not a dimer. An alternative model is the trimeric 53BP1 OD, which results in higher-order oligomerization of 53BP1 with LC8. Mutation of the OD or LC8 binding would then be expected to produce the observed differences in reductions in focus formation in the OD mutant (ODm) and LC8 binding mutant (LC8m). (**B**) LC8-53BP1 binding and accumulation in DNA repair foci. 53BP1/LC8 complex is in exchange between trimeric and bridged species.

bridged by LC8 dimers, they form drastically different complexes, presumably due to their difference in native oligomeric state. The trimeric OD of 53BP1 results in bounded bridging with an upper limit of a dimer-of-trimers, while LCA5 exhibits unbounded bridging in complex with LC8.

Given that all LC8 sites are filled simultaneously and that QT2 appears to be sufficient for bridging, what is then the reason for multivalent sites in 53BP1-LC8 interactions? The binding enhancement from multivalent LC8 interactions is minimal in this case, contrary to what is observed with ASCIZ QT2-4 (*Walker et al., 2023*; *Reardon et al., 2020*) and Nup159, (*Nyarko et al., 2013*; *Gaik et al., 2015*) for example. WT LBD-OD has a $K_d$ of 0.4 μM, while two-site mutants have $K_d$ between 0.4 and 0.7 μM. Either QT1 or QT3 could be absent and make almost no change to the overall affinity of the interaction (QT23-OD $K_d$ = 0.4 μM QT12=OD $K_d$ = 0.5 μM). A model in which LC8 binding stabilizes

bivalent contacts between 53BP1 and modified histones due to the restriction of the conformational ensemble of 53BP1 has been proposed (*Becker et al., 2018*; *Figure 8A*, 53BP1 dimer). Such a binding mode is seen in the dimeric LC8 binding client, RavP (*Figure 8A*), where LC8 binding restricts its conformational ensemble (*Jespersen et al., 2019b*). While there certainly is some binding enhancement of LBD-OD compared to QT2-OD and could possibly be some ordering of the 53BP1 LBD upon binding LC8 multivalently, we propose a different reason for multivalency in 53BP1-LC8 interactions. While QT2 is both necessary and sufficient for bridging, deletion of part of the linker separating LBD and OD results in a primarily bridged complex (*Figure 6*). This implies that the location of the motif relative to the OD contributes to the bridging interaction. We propose that the motif specificity of QT2 and the position of QT1 and QT3 contribute to tuning the affinity of the bridging interaction, optimizing the relative population of trimer and bridged complexes.

## Heterogeneity of oligomeric states regulates 53BP1 focus formation

Currently, 53BP1 is thought of as a dimer that can form higher-order oligomers (*Becker et al., 2018*; *Zgheib et al., 2009*; *Lou et al., 2020*) and LC8 binding will stabilize the 53BP1 OD dimer, resulting in an increase in affinity for bivalent interactions with histones (*Becker et al., 2018*; *Figure 8A*, 53BP1 dimer). Based on nuclear focus formation and chromatin association of 53BP1 OD and LC8 mutants, loss of LC8 binding results in only a modest reduction in chromatin association compared to the WT, while loss of the OD generates 53BP1 that is dimerized only by LC8 but has a significant reduction in chromatin association. Loss of both the OD and QTs abolishes 53BP1 foci almost completely. This model does not explain why the loss of LC8 is not the same as the loss of the OD. Our alternative model proposed in this work includes trimer and bridged 53BP1 oligomers to account for the change in 53BP1 chromatin association because of the higher affinity for histones due to increased avidity (*Figure 8A*, 53BP1 trimer). In this model, the LC8 binding deletion mutant is a trimer, and the OD deletion mutant is a dimer, also explaining the modest and large reduction in 53BP1 foci for these constructs, respectively (*Becker et al., 2018*). Therefore, our trimer and dimer-of-trimers model fully accounts for changes in 53BP1 function resulting from loss of LC8 binding and should provide a more accurate representation of 53BP1-LC8 interactions.

Given that QT2 is essential for bridging, the mutants were designed for testing the specific contribution of the bridged complex. Variants that reduced LC8 binding in QT2 (T1171E, QT3, QT1,3, Null) had the lowest number of foci formation. Important to note is that in our model, bridging improves focus formation, and, therefore, we are looking at small differences, a tuning effect rather than an on/off switch. With QT2 alone, foci formation is restored almost to the level of WT, compared to the Null, givng confidence that this difference is real. The T1171I mutant that does not reduce binding shows foci formation like WT. Taken together, these data support the contribution of bridging to enhancing foci formation and that QT2 binding is critical for this process, either through its specific motif binding or through its location relative to the OD as inferred from the linker deletion mutants, or both. In fact, previous experiments have shown that mutation of only T1171 to alanine results in loss of OD-independent foci formation (*Becker et al., 2018*). The importance of T1171 in QT2 for bridging of 53BP1 trimers provides a possible explanation for the large impact of a single site mutation in 53BP1 foci formation. It is possible that bridging of 53BP1 trimers by LC8 is an important mechanism for regulating 53BP1 oligomeric state and function, but further research is necessary to determine whether this mechanism is physiologically relevant in 53BP1 foci formation. Current data suggest that 53BP1 is a dimer which forms higher-order assemblies under conditions of DNA damage (*Lou et al., 2020*). Future work will determine how the heterogeneous assemblies formed by 53BP1 are linked to its physiological functions.

In summary, we show for the first time that 53BP1 OD is a homogeneous trimer. Binding to LC8 results in a heterogeneous mixture of complexes with masses consistent with the formation of both trimer and bridged complexes of 53BP1, and possibly others. We speculate that heterogeneity in 53BP1-LC8 complexes may allow for improved recruitment of 53BP1 into nuclear repair foci, and the bridged complex stabilizes interactions with chromatin by allowing for more bivalent interactions (*Figure 8A*). Both the trimer and bridged complexes can bind multiple histones, but the dimer-of-trimers could bind up to three histones (blue) bivalently, resulting in high avidity and retention in DNA repair foci (*Figure 8B*). Indeed, mutants deficient in bridging 53BP1 trimers in vitro failed to elicit an improvement in 53BP1 focus formation, while even the weakly bridging QT2 construct improved

focus formation significantly compared to an LC8 binding null mutant, providing support for the physiological relevance of LC8 in bridging 53BP1 trimers (*Figure 6D and E*). Our model (*Figure 8*) explains previous data involving 53BP1-chromatin interactions and offers a new role for LC8 in the enhancement of 53BP1 foci.

## Methods

### Cloning and site-directed mutagenesis

Studies were carried out using human 53BP1 (Uniprot: Q12888) and human LC8-2 (Uniprot: Q96FJ2). Plasmid containing 53BP1 1140–1290 was codon-optimized for expression in *Escherichia coli* and cloned into a pET24d expression vector (GenScript, Piscataway, NJ, USA). All constructs contained an N-terminal hexahistidine tag and tobacco etch virus (TEV) protease cleavage site to facilitate removal of the His-tag. Cysteine residues in the sequence were mutated to serine to avoid disulfide-induced oligomerization. All mutagenesis was performed using New England Biolabs (Ipswich, MA, USA) site-directed mutagenesis kit and custom primers. The WT 53BP1 LBD-OD contains all three LC8 binding sites, while the QT-OD variants contain only the indicated LC8 binding sites i.e. QT1-OD contains intact LC8 binding site 1, while LC8 binding sites 2 and 3 are mutated to AAA to abolish binding. T1171 point mutants are LBD-OD variants containing only a single amino acid change, as indicated in the name of the mutant. Linker deletion mutants are LBD-OD variants with indicated truncation of residues from LBD-OD linker.

### Protein expression and purification

All constructs were transformed into *E. coli* Rosetta DE3 cells and expressed in rich autoinducing media at 37°C for 24 hr or grown with Luria-Bertani and induced at 18°C for 16–20 hr with 1 mM IPTG. For NMR experiments, cells were grown in MJ9 minimal media supplemented with $^{13}$C-glucose and/or $^{15}$NH$_4$Cl. Cells were harvested and purified on Talon His-Tag Purification Resin (Takara Bio, Mountain View, CA, USA). The hexahistidine tag was removed by incubating overnight at 4°C with TEV protease. Cleavage with TEV protease leaves an N-terminal Gly-Ala-His preceding the peptide of interest. The cleaved samples were passed back over Talon resin to remove the cleaved tag and TEV protease. Samples were then purified on S200 Superdex size-exclusion column using an AKTA-FPLC (GE Healthcare, Chicago, IL, USA) to a purity of >95%, as analyzed by SDS-PAGE. Proteins were stored at 4°C and either used within 1 week or flash-frozen and stored at –80°C.

Protein concentration was quantified using absorbance at 280 and 205 nm since the 53BP1 LBD-OD sequence contains only two tyrosine residues using extinction coefficients of: at 205 nm LC8-2 374,720 M$^{-1}$cm$^{-1}$ and LBD-OD constructs (including mutants) 508,340 M$^{-1}$cm$^{-1}$; at 280 nm LC8-2 14440 M$^{-1}$cm$^{-1}$ and LBD-OD constructs (including mutants) 2980 M$^{-1}$cm$^{-1}$.

### Circular dichroism

Spectra were recorded on a JASCO J-810 spectropolarimeter using a 1 mm cell. Protein samples were dialyzed overnight in 2 L of 20 mM sodium phosphate (pH 7.2) prior to data collection. Spectra were collected at room temperature at a protein concentration of 10–15 µM. Data shown are the average of three scans, and results are reported in mean residue molar ellipticity (deg*cm$^2$/dmol).

For thermal denaturing assay, spectra were recorded at the specified temperatures (between 25°C and 80°C). A single sample was used for each assay without replacing the sample between points. Samples were allowed to thermally equilibrate for 10 min prior to data acquisition.

### Isothermal titration calorimetry

Thermodynamics of the 53BP1 LBD-OD:LC8 interaction was measured at 25°C using a VP-ITC microcalorimeter (MicroCal, Northampton, MA, USA). The binding buffer contained 50 mM sodium phosphate and 150 mM sodium chloride (pH 7.2). LC8 at a concentration of 250–400 µM was titrated into an LBD-OD construct. Cell concentrations were between 10 and 35 µM for each LBD-ODm. For experiments where LBD-ODm was titrated into LC8, LBD-OD at a concentration of 300 µM was titrated into LC8 mutant at a concentration of 10–15 µM. Peak areas were integrated, and the data were fit into a one-site binding model in Origin 7.0 (OriginLab Corporation, Northampton, MA, USA). From this, stoichiometry (N), dissociation constant (K$_d$), change in enthalpy (ΔH), and entropy (ΔS)

were determined. Reported data are the average of triplicate runs. Error reported is the standard deviation of the data acquired.

Each isotherm of the one-site mutants of LBD-OD was additionally analyzed using the SBS models found within Origin for VP-ITC. We model that the hexamer binds 3 LC8 dimers, 2 of which bind with identical conditions (intratrimer; between two LBD chains within the same trimer) and 1 binds in a different condition (bridging; linking two 53BP1 trimers together). For SBS, the following constraints were applied: k1=k2; h1=h2; k1>k3; h1<h3. This fitting strategy was followed for every experimental replicate of each one-site mutant, and inverse-variance weighting (*Equations 1 and 2*) was used to aggregate the measured values and their associated errors for each replicate. For a measurement $y_i$ and associated error $\sigma_i$, inverse-variance weighting for calculating an aggregated measurement ($\hat{y}$) and error ($\hat{\sigma}$) can be represented by the following equations:

$$\hat{y} = \frac{\Sigma_i \left( y_i / \sigma_i^2 \right)}{\Sigma_i \left( 1 / \sigma_i^2 \right)} \tag{1}$$

$$\hat{\sigma} = \frac{1}{\Sigma_i \left( 1 / \sigma_i^2 \right)} \tag{2}$$

## Size-exclusion chromatography coupled to multi-angle light scattering

Average molar masses and association states of proteins were determined from SEC (AKTA FPLC; GE Healthcare) coupled to MALS (DAWN; Wyatt Technology) and refractive index (Optilab; Wyatt Technology) detectors. Size-exclusion chromatography was performed on S200 Superdex gel filtration column using an AKTA-FPLC (GE Healthcare, Chicago, IL, USA). Data for free 53BP1 LBD-OD or OD were collected by injecting protein at 1–10 mg/mL with the column pre-equilibrated with the ITC buffer described above. For binding experiments, 50 µM 53BP1 LBD-ODm was mixed with 200 µM LC8 and allowed to bind for 30 min at 4°C. Samples were run at a flow rate of 0.6 mL/min. Average molar masses and associated uncertainties were computed with the ASTRA software package, version 8 (Wyatt Technologies). Expected masses are as follows: 53BP1 LBD-OD 17.0 kDa, LC8 21.2 kDa, OD 13.2 kDa. Protein concentrations were measured using Optilab refractive index detector (Wyatt Technology, Santa Barbara, CA, USA).

## SV-AUC

AUC experiments were performed using a Beckman Coulter Optima XL-A ultracentrifuge equipped with absorbance optics (Brea, CA, USA). For SV-AUC experiments, samples were loaded into Epon two-channel sectored cells with a 12 mm optical pathlength and run at 42,000 rpm in a four-cell Beckman Coulter AN 60-Ti rotor at 20°C. Scans were performed continuously for a total of 300 scans per cell. Data were fit into a continuous c(S) distribution using the software SEDfit. Sedimentation coefficients are expressed in Svedbergs (S).

## NMR

NMR experiments were performed on a Bruker 800 MHz Avance III HD spectrometer (Bruker Biosciences, Billerica, MA, USA) equipped with a 5 mm TCI cryoprobe with Z-axis gradient (Bruker). NMR experiments were carried out at 10°C in 20 mM sodium phosphate, 50 mM sodium chloride, and 1 mM sodium azide (pH 6.5), with 10% $D_2O$, a protease inhibitor mixture (Roche Applied Science, Madison, WI, USA), and 2,2 dimethylsilapentane-5-sulfonic acid for chemical shift referencing. NMR data were processed using nmrPipe (*Delaglio et al., 1995*), and nonuniform sampling artifacts removed using SMILE (*Ying et al., 2017*). Backbone resonances were previously assigned in the LBD (*Howe et al., 2022*) and have been deposited in the BMRB (BMRB entry 51475). Data analysis was performed on NMRViewJ (*Johnson, 2004*). Steady-state $^1$H-$^{15}$N heteronuclear nuclear Overhauser effect ({$^1$H}-$^{15}$N NOE) experiments were collected with an 8 s saturation time. Error bars were calculated using:

$$\sigma/(\mathrm{NOE}) = [(\sigma\ \mathrm{I_{sat}}/\mathrm{I_{sat}})^2 + (\sigma\mathrm{I_{unsat}}/\mathrm{I_{unsat}})^2]^{1/2}$$

where $\mathrm{I_{sat}}$ and $\sigma\mathrm{I_{sat}}$ are the intensity of the peak and its baseline noise, respectively. R1 and R2 data were collected with HSQC-based temperature-compensated pulse sequences (*Farrow et al., 1994*).

Time points were collected in triplicate for error estimation. For titration of LBD-OD with LC8, $^1$H-$^{15}$N HSQC spectra of 100 µM $^{15}$N-labeled LBD-OD were collected with 0.25, 0.5, 1, 2, 3, and 4 molar equivalents of unlabeled LC8 titrated using the NMR conditions described above. Chemical shift indexing was performed using sequence-corrected shifts for amino acids in random coils (*Kjaergaard and Poulsen, 2011*; *Schwarzinger et al., 2001*; *Kjaergaard et al., 2011*).

## Immunofluorescence staining

U2OS cells were transfected with 3 µg GFP expression vectors using 10 µL polyethylenimine (1 mg/mL). Cells were irradiated using MultiRad (Precision XRay Irradiation) with indicated dose 24 hr after transfection followed by 1 hr rest. The coverslips were then fixed with 3% paraformaldehyde for 10 min and permeabilized using 0.5% Triton X-100 solution for 5 min. Samples were incubated with mouse monoclonal primary gH2AX antibody (EMD Millipore, 05-636) (1:1000) for 1 hr followed by Alexa Fluor 594 goat anti-mouse (1:500) and DAPI (1:5000) for 30 min. Coverslips were mounted using 0.02% anti-fade solution (0.02% in 90% glycerol). Samples were imaged using Zeiss LSM900 confocal microscope and analyzed using GraphPad Prism. Data represent three to four individual replicates.

## Acknowledgements

The authors would like to thank Austin Weeks for help generating reagents. We also thank Nikolaus Loening, Sanjay Ramprasad, and Sarah McGee for thoughtful conversation and editing. JH acknowledges funding from ARCS Oregon Chapter and the Dean's Catalyst award. This work is funded by the National Institutes of Health (R01GM141733 to EB). We also acknowledge the support of the Oregon State University NMR Facility funded by the National Institutes of Health, HEI grant 1S10OD018518, and the MJ Murdock Charitable Trust grant #2014162. JWL is supported by a grant from NIH (NIGMS: R35GM137798, NCI: R01CA244261), American Cancer Society (RSG-20-131-01-DMC and TLC-21-164-01-TLC), and University of Texas STARs award.

## Additional information

### Funding

| Funder | Grant reference number | Author |
| --- | --- | --- |
| National Institute of General Medical Sciences | R01GM141733 | Elisar J Barbar |
| National Institute of General Medical Sciences | R35GM137798 | Justin WC Leung |
| National Cancer Institute | R01CA244261 | Justin WC Leung |
| American Cancer Society | 10.53354/acs.rsg-20-131-01-dmc.pc.gr.142479 | Justin WC Leung |
| ARCS Oregon Chapter | Graduate Student Fellowship | Jesse Howe |
| National Institutes of Health | 1S10OD018518 | Elisar J Barbar |
| American Cancer Society | 10.53354/pc.gr.151537 | Justin WC Leung |
| MJ Murdock Charitable Trust | #2014162 | Elisar J Barbar |
| University of Texas Health Science Center at San Antonio | STARs award | Justin WC Leung |

The funders had no role in study design, data collection and interpretation, or the decision to submit the work for publication.

## Author contributions
Jesse Howe, Conceptualization, Formal analysis, Investigation, Methodology, Writing – original draft, Writing – review and editing; Douglas Walker, Formal analysis, Writing – review and editing; Kyle Tengler, Maya Sonpatki, Investigation; Patrick N Reardon, Formal analysis, Investigation, Writing – review and editing; Justin WC Leung, Formal analysis, Funding acquisition, Investigation, Writing – original draft, Writing – review and editing; Elisar J Barbar, Conceptualization, Supervision, Funding acquisition, Investigation, Project administration, Writing – review and editing

## Author ORCIDs
Justin WC Leung ⓘD https://orcid.org/0000-0003-4601-390X
Elisar J Barbar ⓘD https://orcid.org/0000-0003-4892-5259

## Decision letter and Author response
Decision letter https://doi.org/10.7554/eLife.102179.sa1
Author response https://doi.org/10.7554/eLife.102179.sa2

---

# Additional files

## Supplementary files
MDAR checklist

## Data availability
All data generated or analysed during this study are included in the manuscript; source data files have been provided for Figures 1–4, 6 and 7.

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
