## [Editor Report]

This study offers a useful investigation into how 53BP1 and LC8 interact to form higher-order oligomers that are important for DNA repair. The authors provide convincing biochemical and biophysical evidence supporting a model in which LC8 bridges 53BP1 trimers via the QT2 motif. The work establishes a solid foundation for future efforts aimed at elucidating the structural organization and functional relevance of these complexes in vivo, and will be of broad interest to researchers studying DNA damage response and protein complex assembly.

---

## [Decision Letter]

**Decision letter after peer review:**

Thank you for submitting your article "LC8 enhances 53BP1 foci through heterogeneous bridging of 53BP1 oligomers" for consideration by *eLife*. Your article has been reviewed by 2 peer reviewers, and the evaluation has been overseen by a Reviewing Editor and Amy Andreotti as the Senior Editor. The reviewers have opted to remain anonymous.

Essential Revisions:

1) Provide additional evidence for the trimeric model of 53BP1 oligomers:

The current conclusion that 53BP1 forms stable trimers is primarily based on SEC-MALS data, which are limited by protein heterogeneity and elongated shapes. The authors should consider alternative methods (e.g., cross-linking, additional biophysical techniques) to validate the proposed trimeric state or acknowledge that this remains speculative (cf. Reviewer 1).

2) Reassess the "dimer-of-trimers" model for LC8-mediated bridging:

The SEC-MALS data near the void volume result in imprecise molecular weight estimates, leaving open the possibility of alternative stoichiometries (e.g., dimers or higher-order assemblies). Clarify the limitations of the current data or explore additional experiments to strengthen this claim (cf. Reviewer 1)

3. Address uncertainties in stoichiometry and oligomeric transitions:

The coexistence of multiple oligomeric states (dimers, trimers, and higher-order forms) in solution raises questions about the proposed assembly pathway. Consider computational modeling, mass spectrometry, or additional mutational analysis to clarify how these transitions occur and their relevance to DNA repair (cf. Reviewer 1)

4. Test the physiological relevance of LC8-mediated 53BP1 oligomerization:

While the in vitro data suggest a novel mechanism, the study does not sufficiently demonstrate the importance of LC8-mediated trimer bridging in living cells. Complementary cellular experiments or a discussion acknowledging this limitation would be beneficial (cf. Reviewer 1 and 2)

*Reviewer #1 (Recommendations for the authors):*

This study presents a useful investigation into how 53BP1 protein oligomers may be organised upon engaging chromatin at sites of DNA damage. The work concentrates on the oligomeric state of 53BP1's intrinsic oligomerization domain (OD; as yet incompletely resolved in the literature), and its interplay with LC8 dimers, that by binding QT motifs in OD-proximal upstream disordered domain can further modulate 53BP1 multimeric state and/or the organisation of oligomers.

Here, the data apply biophysical approaches including SEC-MALS, NMR and ITC to better understanding of 53BP1 oligomeric state Â{plus minus}LC8-mediated dimerisation of the 53BP1 QT motifs. Resulting data is suggestive of higher order 53BP1 oligomers, formed of OD-mediated trimers that can be further bridged/linked by LC8-dependent QT motif dimerisation. The evidence hints at such an activity, which if correct, would advance our understanding of how chromatin reader proteins regulate DNA repair. However, there are limitations to the biophysical analyses and the validity of their interpretation, which calls into question the certainty of some of the central conclusions.

Summary

Howe et al. explore the interaction between 53BP1 and LC8 using various techniques, including analytical ultracentrifugation, isothermal titration calorimetry, and size-exclusion chromatography. The authors determine that (1) the oligomerization domain (OD) of 53BP1 exists as a trimer, which is unusual for LC8 clients, which typically form dimers or tetramers. They note that LC8 promotes the assembly of a complex consisting of a dimer-of-trimers, thereby enhancing the localization of 53BP1 at DNA damage sites. (2) The second of the three LC8 recognition motifs in 53BP1 is crucial for stable complex formation, Overall, these results challenge the current view of 53BP1 as exclusively dimeric, emphasizing the importance of trimerization and heterogeneous bridging in the formation of 53BP1 foci.

Strengths/weaknesses/limitations – focusing on main messages:

1. Proposed trimeric model for 53BP1 oligomers – There is still limited evidence to substantiate the proposed trimeric form of 53BP1. The authors' hypothesis of this unusual oligomeric state is based on SEC-MALS experiments. Notably, the proteins 53BP1 LBD and LBD-OD do not exhibit globular structures; instead, they have disordered and elongated shapes. Additionally, as noted by Omar Zgheib et al., higher oligomeric states of 53BP1, such as tetramers, coexist with dimers and quickly interconvert in solution, which can lead to mass heterogeneity in SEC peaks, as evidenced in figures 1F and 1H showing slopes across the peaks. In such cases, the trimeric form of 53BP1 might represent an intermediate state during the transition from dimer to tetramer.

2. Proposed model of LC8-dependent bridging of 53BP1 oligomers – There is a lack of direct evidence supporting its classification as a "dimer of trimers", given that the SEC-MALS peaks are very close to the void volume, leading to inaccurate molecular weight estimates. Since 53BP1 is known to form higher-order oligomers, it is plausible that dimer-of-dimers with six LC8 dimers (MW 188 kDa) or trimer-of-dimers (hexamer) with an MW of 282 kDa could also align with the estimated molecular weights from SEC-MALS in this study.

Although the isothermal titration calorimetry (ITC) experiments in Figure 5A suggest that LC8 dimers interact with the 53BP1 LBD-OD at a 3:1 stoichiometry, this could also supports an alternative model where three LC8s bind to a 53BP1 LBD-OD dimer.

3. Relevance of proposed LC8-dependent bridging of 53BP1 – a weakness is the work does not go far enough to establish and/or test the physiological relevance of the hypothesized 53BP1 trimer bridging, or the function of 53BP1-LC8 interplay, beyond what was already known. LC8-enforced 53BP1 dynamics within foci at DNA damage sites was already experimentally demonstrated in the literature, using more precise measurements in living cells where more information could be gleaned.

Points to consider and/or address:

1) Figure 3 – 3B Again the SEC peak was over interpreted. It is too close to the void, hence the mass result was influenced by the aggregates.

2) Figure 4 – The AUC data need to process further – calculate approximate MW of the complex

3) The X axis labels in panel C do not align very well. The peak with intermedium MW in LBD-OB, between the "bridge" complex and "intramolecular" complex has not been assigned to any model.

4) Figure 5. Panel L: A sequential binding model may not be suitable to fit the data in LBD and mutants except QT3 and QT13 which showed obvious biphasic binding curves.

5) Figure 6. Panel B the peak assigned to "bridge complex" seems to have a ~400kDa mass. Addition information/evidence is necessary to demonstrate the bridge complex exists.

6) Figure 7. Panel D and its respective quantification in F, do not seem to correlate. The thresholding scoring '% cells with >10 foci' is also not appropriate for this assay, since the defect is unlikely be numeric (i.e. affect foci number), given that the LC8 motifs are not essential for recruitment into DNA damage foci in the context of such a minimal protein fragment. Focal intensity, and protein dynamics/mobility in foci, however, are much more likely be affected by loss of LC8 bridging: e.g. altered dynamics/residence time of QT mutant 53BP1 was previously demonstrated by FRAP in Ref 6. The data in 7D-E do not capture such altered behavior, and conversely offer less useful insight than the previous work.

*Reviewer #2 (Recommendations for the authors):*

The authors characterize mixed oligomers formed by the DNA repair factor 53BP1 and the scaffold protein LC8 (dynein light chain 8). Using biochemistry and biophysical experiments they show that various mixed heterooligomers can form, involving OD trimers and LC8 dimers with varying stoichiometries, depending on specific interaction LC8 binding motifs present in an 53BP1 IDR.

The manuscript presents interesting and significant findings on the 53BP1/LC8 interactions, supported by strong biochemical and biophysical evidence, which is linked to the formation of 53BP1 foci in cells.

The authors study the formation of mixed oligomers involving the oligomerization domain (OD) of the DNA repair factor 53BP1 and the scaffold protein LC8 (dynein light chain 8). Using biochemistry and biophysical experiments they show that various mixed heterooligomers are formed involving OD trimers and LC8 dimers with varying stoichiometries, depending on specific interaction motifs (3 LC8 binding motifs "QT") in an 53BP1 IDR and affinities. By mutational analysis in vitro and in cell experiments an important role for the formation of heterooligomers is found for the second motif, QT2. The authors propose that a novel binding mode for LC8, i.e. bridging of trimeric client complexes may play a role for 53BP1 foci formation.

The manuscript presents a careful and interesting analysis of molecular features that contribute to the formation of specific higher order complexes from 53BP1 trimers and LC8 dimers, which bridge two 53BP1 trimers, mainly depending on QT2 in the 53BP1 binding region for LC8.

This is interesting study will be useful to dissect the specific roles and mechanisms of LC8 oligomers with 53BP1 and link them with biological function. The work is technically sound and well written, considering the complexity of the systems. Structural details and mechanisms for the proposed oligomerization states are not provided, but obtaining these will also be highly challenging and laborious. Nevertheless, the study sheds new light onto mechanisms of LC8 mediated oligomerization with unique features with 53BP1 and a specific role in DNA repair.

Specific comments:

– The role of the oligomerization for foci information is an interesting aspect. However, the cellular data are more complex than the binary in vitro analysis with just the two proteins. This should be clarified. E.g. what is required for foci formation in cells, are the additional factors that could bind to either of the proteins that need to be considered?

– Can the authors reconstitute droplets in vitro from mixing the two proteins with additional components, i.e. full-length proteins and/or DNA?

– For the condensate formation one would expect multivalent interactions to play a role. While this is clearly possible given the oligomerization of the two components a defined 2:3 heteroligomer would not necessarily enable the formation of large networks in a condensate state. Can this be explored with the QT mutants using in vitro condensate assays?

– Figure 2A: the authors should comment on the lack of signals seen for the OD. Have they tried to observe signals for the OD at higher temperature and using deuterated protein?

– Figure 1 panel labels/explanations are not correct, i.e. for (F).

– Please provide in each figure legend, which constructs have been used (residue numbers and domains included) for the respective data shown for LC8 and 53BP1. This seems not always consistent, e.g. in Figure 1E 1200-1290 is used, which in Figure 1A LBD-OD is indicated as 1140-1290, in Figure 2, OD signals are apparently missing, make a comment on this in the legend.

– How can the ITC data be rationalized, e.g. the different titration curves and the different contributions of enthalpy/entropy for QT motifs. Can this be rationalized with structural models of the QT2 motifs binding to LC8? Why does QT2 have unfavourable binding entropy distinct from QT1 and QT3? And why is the QT13-OD non sigmoidal?

---

## [Author Response]

Essential Revisions:1) Provide additional evidence for the trimeric model of 53BP1 oligomers:The current conclusion that 53BP1 forms stable trimers is primarily based on SEC-MALS data, which are limited by protein heterogeneity and elongated shapes. The authors should consider alternative methods (e.g., cross-linking, additional biophysical techniques) to validate the proposed trimeric state or acknowledge that this remains speculative (cf. Reviewer 1).

While SEC mass estimation is dependent on conformation, MALS is not. The use of SEC coupled to MALS is to separate components of heterogeneous systems to be measured separately. Our mass measurement by MALS is therefore affected by peak overlap, but not elongated shape. We are careful to measure mass by MALS near the center of peaks, where most of the sample is available to scatter and there is minimal peak overlap. It is our understanding that this is common practice for MALS measurements.

We would like to emphasize the large loading concentration range (75-750 μM) in which the mass measurement of the OD is constant, suggesting a lack of concentration-dependent exchange between oligomeric states (i.e. Absence of dimer/tetramer exchange). The Figure 1 legend was missing this description, and this is possibly why the reviewers missed it.

Another evidence that strongly supports the trimer is that we have done SEC-MALS on a smaller construct containing the OD, and that also showed a mass for a trimer (Figure 1). Having two different size constructs showing masses for a trimer gives more confidence in the reproducibility of the results.

Alternative experiments for measuring the mass of 53BP1 OD accurately were attempted. AUC sedimentation equilibrium experiments were run several times, but the extended length of the experiment resulted in significant degradation of the sample and inaccurate mass estimates. Native mass spectrometry experiments were attempted and showed primarily a monomer peak as expected because a monomer is more easily observed in the gas phase. It also showed a minor peak for a trimer and a hexamer and therefore while inconclusive, are consistent with a trimeric structure over a dimeric or tetrameric.

We have included in the Discussion (see tracked version) a summary of all the evidence for a trimer.

2) Reassess the "dimer-of-trimers" model for LC8-mediated bridging:The SEC-MALS data near the void volume result in imprecise molecular weight estimates, leaving open the possibility of alternative stoichiometries (e.g., dimers or higher-order assemblies). Clarify the limitations of the current data or explore additional experiments to strengthen this claim (cf. Reviewer 1)

While SEC-MALS data are near the void volume, as discussed above MALS data are not dependent on the void volume as it can be determined even without a colum. The value of the column is to separate these peaks for better mass determination, but the highest mass determined is not affected by void volume.

We have included in this resubmission a paragraph in the Discussion that summarizes all the data that point to a dimer of trimer model, and also where appropriate we are less assertive in proposing this model as in the absence of structure it is very difficult to be overly confident that this is the only possible model.

It is possible for a dimer and tetramers to exist, but that we do not see any evidence for them.

3. Address uncertainties in stoichiometry and oligomeric transitions:The coexistence of multiple oligomeric states (dimers, trimers, and higher-order forms) in solution raises questions about the proposed assembly pathway. Consider computational modeling, mass spectrometry, or additional mutational analysis to clarify how these transitions occur and their relevance to DNA repair (cf. Reviewer 1)

As discussed above, mass spectrometry of the unbound protein is consistent with a primarily trimeric and hexameric populations (data not shown). We were not successful in seeing anything in the bound. We have tried modeling using AlphaFold but that did not produce any reasonable structures. Instead we have made several variants and analyzed their binding by several methods. Collectively they are consistent with the pathway proposed, although in the absence of structures, the pathway is still a proposal.

Introducing DNA repair experiments is beyond the scope of this study.

4. Test the physiological relevance of LC8-mediated 53BP1 oligomerization:While the in vitro data suggest a novel mechanism, the study does not sufficiently demonstrate the importance of LC8-mediated trimer bridging in living cells. Complementary cellular experiments or a discussion acknowledging this limitation would be beneficial (cf. Reviewer 1 and 2)

We have attempted to clarify the discussion of the importance of LC8 bridging by comparing mutants that specifically affect the bridged complex. We have added this paragraph:

“Given that QT2 is essential for bridging, the mutants were designed with testing the specific contribution of the bridged complex. Variants that reduced LC8 binding in QT2 (T1171E, QT3, QT1,3, Null) had the lowest number of foci formation. Important to note is that in our model, bridging improves focus formation and therefore we are looking at small differences, a tuning effect rather than an on/off switch. With QT2 alone restoring foci formation almost to the level of WT, compared to the Null, gives confidence that this difference is real. The T1171I mutant that does not reduce binding show similar foci formation like WT. Taken together, these data support contribution of bridging to enhancing foci formation, and that QT2 binding is critical for this process, either through its specific motif binding or through its location relative to the OD as inferred from the linker deletion mutants, or both.”

Reviewer #1 (Recommendations for the authors):This study presents a useful investigation into how 53BP1 protein oligomers may be organised upon engaging chromatin at sites of DNA damage. The work concentrates on the oligomeric state of 53BP1's intrinsic oligomerization domain (OD; as yet incompletely resolved in the literature), and its interplay with LC8 dimers, that by binding QT motifs in OD-proximal upstream disordered domain can further modulate 53BP1 multimeric state and/or the organisation of oligomers.Here, the data apply biophysical approaches including SEC-MALS, NMR and ITC to better understanding of 53BP1 oligomeric state Â{plus minus}LC8-mediated dimerisation of the 53BP1 QT motifs. Resulting data is suggestive of higher order 53BP1 oligomers, formed of OD-mediated trimers that can be further bridged/linked by LC8-dependent QT motif dimerisation. The evidence hints at such an activity, which if correct, would advance our understanding of how chromatin reader proteins regulate DNA repair. However, there are limitations to the biophysical analyses and the validity of their interpretation, which calls into question the certainty of some of the central conclusions.SummaryHowe et al. explore the interaction between 53BP1 and LC8 using various techniques, including analytical ultracentrifugation, isothermal titration calorimetry, and size-exclusion chromatography. The authors determine that (1) the oligomerization domain (OD) of 53BP1 exists as a trimer, which is unusual for LC8 clients, which typically form dimers or tetramers. They note that LC8 promotes the assembly of a complex consisting of a dimer-of-trimers, thereby enhancing the localization of 53BP1 at DNA damage sites. (2) The second of the three LC8 recognition motifs in 53BP1 is crucial for stable complex formation, Overall, these results challenge the current view of 53BP1 as exclusively dimeric, emphasizing the importance of trimerization and heterogeneous bridging in the formation of 53BP1 foci.Strengths/weaknesses/limitations – focusing on main messages:1. Proposed trimeric model for 53BP1 oligomers – There is still limited evidence to substantiate the proposed trimeric form of 53BP1. The authors' hypothesis of this unusual oligomeric state is based on SEC-MALS experiments. Notably, the proteins 53BP1 LBD and LBD-OD do not exhibit globular structures; instead, they have disordered and elongated shapes. Additionally, as noted by Omar Zgheib et al., higher oligomeric states of 53BP1, such as tetramers, coexist with dimers and quickly interconvert in solution, which can lead to mass heterogeneity in SEC peaks, as evidenced in figures 1F and 1H showing slopes across the peaks. In such cases, the trimeric form of 53BP1 might represent an intermediate state during the transition from dimer to tetramer.

As mentioned above, MALS is not affected by shape, and is the perfect method for measuring mass of elongated proteins. We obtain the same mass even for the OD alone which is not elongated.

Regarding the Zgheib et al. paper, from what we can understand, they only use SEC in this paper.

We disagree about the trimer being a transition from the tetramer to dimer because we obtain the same mass and the same peak shape in a concentration range that differs by 10-fold (Figure 1F).

2. Proposed model of LC8-dependent bridging of 53BP1 oligomers – There is a lack of direct evidence supporting its classification as a "dimer of trimers", given that the SEC-MALS peaks are very close to the void volume, leading to inaccurate molecular weight estimates. Since 53BP1 is known to form higher-order oligomers, it is plausible that dimer-of-dimers with six LC8 dimers (MW 188 kDa) or trimer-of-dimers (hexamer) with an MW of 282 kDa could also align with the estimated molecular weights from SEC-MALS in this study.Although the isothermal titration calorimetry (ITC) experiments in Figure 5A suggest that LC8 dimers interact with the 53BP1 LBD-OD at a 3:1 stoichiometry, this could also supports an alternative model where three LC8s bind to a 53BP1 LBD-OD dimer.

We have thought very hard about alternative models and it is likely that any of these could be correct. However, the overwhelming evidence that we have is consistent with a dimer of trimers. We have added this paragraph below, and in many places throughout the manuscript we indicated that this is a proposal and there could be other models.

The wild-type LBD-OD (Figure 5A) shows a stoichiometry near 3:1 (3 LC8: 1 53BP1; or 3 LC8 dimers and one dimer of 53BP1; or 9 LC8 dimers and two trimers of 53BP1). The latter suggests that a bridged complex could be formed when all the LC8 sites are filled. However, ITC alone is not sufficient to differentiate between all possible models.

3. Relevance of proposed LC8-dependent bridging of 53BP1 – a weakness is the work does not go far enough to establish and/or test the physiological relevance of the hypothesized 53BP1 trimer bridging, or the function of 53BP1-LC8 interplay, beyond what was already known. LC8-enforced 53BP1 dynamics within foci at DNA damage sites was already experimentally demonstrated in the literature, using more precise measurements in living cells where more information could be gleaned.

We agree that the foci at DNA damage sites was already established, and our work explains the structural basis for this.

We add the following to the Discussion

“In fact, previous experiments have shown that mutation of only T1171 to alanine result in loss of oligomerization domain independent foci formation^6^. The importance of T1171 in QT2 for bridging of 53BP1 trimers provides a possible explanation for the large impact of a single site mutation in 53BP1 foci formation. It is possible that bridging of 53BP1 trimers by LC8 is an important mechanism for regulating 53BP1 oligomeric state and function, but further research is necessary to determine whether this mechanism a physiologically relevant mechanism in 53BP1 foci formation. Current data suggest that 53BP1 is a dimer which forms higher-order assemblies under conditions of DNA damage^27^. Future work will determine how the heterogeneous assemblies formed by 53BP1 are linked to its physiological functions.”

Points to consider and/or address:1) Figure 3 – 3B Again the SEC peak was over interpreted. It is too close to the void, hence the mass result was influenced by the aggregates.

This is not entirely true, as it is 1 ml away from the void, and samples were injected right after mixing with no delay time for aggregation. No major aggregation peaks were observed in the AUC chromatograms.

(2) Figure 4 – The AUC data need to process further – calculate approximate MW of the complex.

From the sedimentation velocity we cannot accurately determined the MW of the complex because of the disorder.

3) The X axis labels in panel C do not align very well. The peak with intermedium MW in LBD-OB, between the "bridge" complex and "intramolecular" complex has not been assigned to any model.

The peak with intermediate MW is for an intermediate that can be partially occupied with LC8 as shown in Figure 3D.

4) Figure 5. Panel L: A sequential binding model may not be suitable to fit the data in LBD and mutants except QT3 and QT13 which showed obvious biphasic binding curves.

The reviewer is correct, a sequential model is not suitable and that’s why did analysis with both methods for all binding interactions.

5) Figure 6. Panel B the peak assigned to "bridge complex" seems to have a ~400kDa mass. Addition information/evidence is necessary to demonstrate the bridge complex exists.

The mass starting with about 300 is the same as that of Figure 3B. What is important in this data is that there is a major shift to a higher complex, by changing the length of the linker one obvious explanation is an effect on bridging, rather than binding and this data support the bridged complex.

6) Figure 7. Panel D and its respective quantification in F, do not seem to correlate. The thresholding scoring '% cells with >10 foci' is also not appropriate for this assay, since the defect is unlikely be numeric (i.e. affect foci number), given that the LC8 motifs are not essential for recruitment into DNA damage foci in the context of such a minimal protein fragment. Focal intensity, and protein dynamics/mobility in foci, however, are much more likely be affected by loss of LC8 bridging: e.g. altered dynamics/residence time of QT mutant 53BP1 was previously demonstrated by FRAP in Ref 6. The data in 7D-E do not capture such altered behavior, and conversely offer less useful insight than the previous work.

We agree with the reviewer that FRAP would have been a better experiment.

Reviewer #2 (Recommendations for the authors):The authors characterize mixed oligomers formed by the DNA repair factor 53BP1 and the scaffold protein LC8 (dynein light chain 8). Using biochemistry and biophysical experiments they show that various mixed heterooligomers can form, involving OD trimers and LC8 dimers with varying stoichiometries, depending on specific interaction LC8 binding motifs present in an 53BP1 IDR.The manuscript presents interesting and significant findings on the 53BP1/LC8 interactions, supported by strong biochemical and biophysical evidence, which is linked to the formation of 53BP1 foci in cells.The authors study the formation of mixed oligomers involving the oligomerization domain (OD) of the DNA repair factor 53BP1 and the scaffold protein LC8 (dynein light chain 8). Using biochemistry and biophysical experiments they show that various mixed heterooligomers are formed involving OD trimers and LC8 dimers with varying stoichiometries, depending on specific interaction motifs (3 LC8 binding motifs "QT") in an 53BP1 IDR and affinities. By mutational analysis in vitro and in cell experiments an important role for the formation of heterooligomers is found for the second motif, QT2. The authors propose that a novel binding mode for LC8, i.e. bridging of trimeric client complexes may play a role for 53BP1 foci formation.The manuscript presents a careful and interesting analysis of molecular features that contribute to the formation of specific higher order complexes from 53BP1 trimers and LC8 dimers, which bridge two 53BP1 trimers, mainly depending on QT2 in the 53BP1 binding region for LC8.This is interesting study will be useful to dissect the specific roles and mechanisms of LC8 oligomers with 53BP1 and link them with biological function. The work is technically sound and well written, considering the complexity of the systems. Structural details and mechanisms for the proposed oligomerization states are not provided, but obtaining these will also be highly challenging and laborious. Nevertheless, the study sheds new light onto mechanisms of LC8 mediated oligomerization with unique features with 53BP1 and a specific role in DNA repair.Specific comments:– The role of the oligomerization for foci information is an interesting aspect. However, the cellular data are more complex than the binary in vitro analysis with just the two proteins. This should be clarified. E.g. what is required for foci formation in cells, are the additional factors that could bind to either of the proteins that need to be considered?

We agree with the reviewer that the cellular data are much more complex and there are many other players as nicely explained in the introduction. However, this is the best we can do as a footnote in this largely biophysical study and the results we see while do not tell the whole story, show the small difference that is consistent with our model developed from in vitro studies.

– Can the authors reconstitute droplets in vitro from mixing the two proteins with additional components, i.e. full-length proteins and/or DNA?

This experiment is a good idea but it is way beyond the scope of this work.

– For the condensate formation one would expect multivalent interactions to play a role. While this is clearly possible given the oligomerization of the two components a defined 2:3 heteroligomer would not necessarily enable the formation of large networks in a condensate state. Can this be explored with the QT mutants using in vitro condensate assays?

This is a good idea, but we did not see any evidence for it, and finding conditions is beyond the scope of this work.

– Figure 2A: the authors should comment on the lack of signals seen for the OD. Have they tried to observe signals for the OD at higher temperature and using deuterated protein?

We have tried changing the temperature up to 40C, but we did not try deuteration.

– Figure 1 panel labels/explanations are not correct, i.e. for (F).

This has been corrected, thank you

– Please provide in each figure legend, which constructs have been used (residue numbers and domains included) for the respective data shown for LC8 and 53BP1. This seems not always consistent, e.g. in Figure 1E 1200-1290 is used, which in Figure 1A LBD-OD is indicated as 1140-1290, in Figure 2, OD signals are apparently missing, make a comment on this in the legend.

Thank you for this comment, we have included this information in the revised version.

– How can the ITC data be rationalized, e.g. the different titration curves and the different contributions of enthalpy/entropy for QT motifs. Can this be rationalized with structural models of the QT2 motifs binding to LC8? Why does QT2 have unfavourable binding entropy distinct from QT1 and QT3? And why is the QT13-OD non sigmoidal?

We can only speculate, QT2 is the only site with an unfavorable entropy and very high enthalpic contribution. Also the placement of QT2 in the correct position relative to the OD is a factor, as suggested by the increase in bridging observed in the linker mutants. Additionally, the fact that QT2-OD has the strongest affinity for the bridging interaction provides a mechanistic explanation for our observation that QT2 is essential for a strongly bridged dimer-of-trimers complex.

We have added the following:

“Since all LBD-OD mutants containing QT2 have sigmoidal isotherms, we attribute the non-sigmoidal isotherms to the formation of a trimeric intermediate and suggest that QT2 binding may pay the entropic penalty for bridging 53BP1 trimers.”